

# Quantification of Baltic Sea Water Budget components Using Dynamic Topography

Vahidreza Jahanmard[1], Artu Ellmann[1], Nicole Delpeche-Ellmann[2]

[1] Department of Civil Engineering and Architecture, Tallinn University of Technology, Ehitajate tee 5, Tallinn, 19086, Estonia

[2] Department of Cybernetics, School of Science, Tallinn University of Technology, Ehitajate tee 5, Tallinn, 19086, Estonia

*Correspondence to*: Vahidreza Jahanmard (vahidreza.jahanmard@taltech.ee)

**Abstract.** Accurate quantification of the Baltic Sea water budget components is essential for understanding both seasonal and long-term variations influenced by climate change. In this study, we utilize dynamic topography (DT), referenced to the geoid, to derive dynamic water volume and improve estimates of the main water balance components, such as river runoff and water exchange through the Danish Straits. We utilize DT for 2017–2021.5, which was corrected for vertical sea level biases and whose vertical datum thus coincides with the geoid. Our findings reveal seasonal dynamic volume variations, with minimum in spring ($78.9 \pm 60$ km$^3$) and maximum in autumn and winter ($121 \pm 57$ km$^3$ and $124 \pm 80$ km$^3$, respectively). Anomalies in DT highlight a specific region (northern Baltic Proper) as representing equilibrium mean DT for the entire Baltic Sea, while areas in the eastern and southern Baltic are prone to extremes. Barotropic exchange analysis shows that no Major Baltic Inflows occurred during the study period, with small to medium inflows averaging 1.6 km$^3$/day in autumn and winter, while outflows averaged 2.36 km$^3$/day. River discharge, indirectly calculated from the water budget, peaked in summer (2.08 km$^3$/day) and was lowest in autumn (1.26 km$^3$/day), with hydrological models underestimating flows in these seasons. As a result, the method and results show great potential for quantification, validation, and a better understanding of the dynamics of the Baltic Sea, especially with a changing climate.

## 1 Introduction

The Baltic Sea is a shallow, semi-enclosed estuary located in northern Europe (Figure 1), that is quite sensitive to present and future impacts of climate change, for instance increasing water temperature and sea level and decreasing ice extent. These impacts are projected to be more pronounced in the Baltic Sea than in the global ocean (HELCOM, 2021; Barghorn et al., 2023; BACC II Author Team, 2015). Therefore, these changes signal the importance of understanding and accurately quantifying some of the main components of the Baltic Sea water budget, that happens to be related to sea level variation of the Baltic Sea.

The water structure of the Baltic Sea is highly stratified with a permanent halocline. This stratification results from several factors: (i) voluminous river runoff, with the largest discharges originating from the northern and eastern sections; (ii) restricted and intermittent saline water inflows from the North Sea, through the narrow and shallow Danish Strait; and (iii)





limited vertical mixing, as convection, mechanical mixing, entrainment, and advection are known to occur for tidal amplitudes are more or less small. In addition, a permanent horizontal density gradient exists in the north to south direction, making the sea level always higher in the northern than southern regions (Leppäranta and Myrberg, 2009).

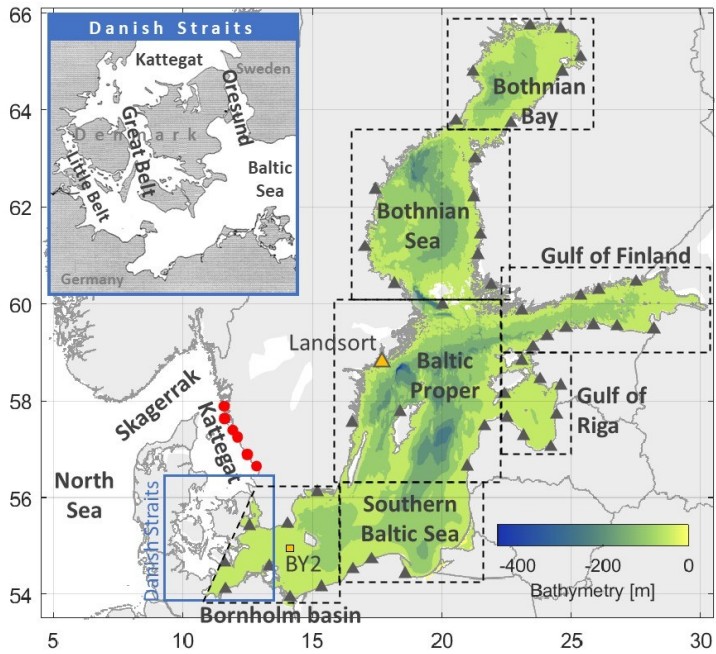

**Figure 1: Location of the Baltic Sea and its depth distribution (obtained from GEBCO 2022). Dashed boxes show the boundaries of the Baltic sub-basins used in this study. Red dots show the location of tide gauges used to calculate volume transport between the Baltic and North Sea. Black triangles denote tide gauge stations that are used for correcting and evaluating the original Nemo-Nordic model in Jahanmard et al. (2023a).**

The direction taken in this study, which to our knowledge has not been previously undertaken, utilizes accurate sea level variation that refers to the geoid (i.e., equipotential surface of the earth), which is also referred to as dynamic topography (DT) (Jahanmard et al., 2021; Rajabi-Kiasari et al., 2023). This approach provides innovative opportunities to quantify the components of the Baltic Sea water budget, and these results can reflect both short-term changes and even climate related links.

In fact, the Baltic Sea water budget is essentially governed by river runoff, evaporation, precipitation and inflow/outflow through the Danish Straits. Therefore, it can be explained by the dynamic water volume variability, which represents the deviation of volume from the constant baseline volume of the basin (volume below the geoid surface). The importance of accurately quantifying DT (and thus the dynamic water volume) is that several studies have suggested that annual and seasonal shifts in Baltic Sea water budget components (e.g. Baltic inflow/outflow, river discharge, sea ice) may overall affect

the whole marine system with serious implications (Barghorn et al., 2023; BACC II Author Team, 2015; Raudsepp et al.,



2023). This signifies the need for an alternative method that allows better quantification of the Baltic Sea water budget components on annual and seasonal time scales. Note also, since the Baltic Sea is divided into several sub-basins (based on its bathymetry and geomorphology) the magnitude and frequency of changes are expected to have a spatial context, with some regions being more affected than others.

Previously, determination of accurate DT for both coastal and offshore areas has been limited due to access of an accurate high-resolution geoid model. The Baltic Sea countries however have developed the NKG 2016 geoid model, whereas presently the geoid based Baltic Sea Chart Datum BSCD2000 vertical datum (Liebsch et al., 2023) is being implemented in the Baltic Sea countries. Adopting the geoid model is suitable to transfer all the sea level datasets to the same zero level that is long-term mean sea level marked at Normaal Amsterdams Peil (NAP), allowing thus seamless sea level data (e.g.

Hydrodynamic Model, tide gauge, and satellite altimetry) integration from coast and offshore.

Ideally, the sea level component in hydrodynamic models (HDM) can be employed for assessing the water budget. However, they are limited by modelling errors and a vertical reference bias that constitutes altogether as an overall bias which varies both spatially and temporally. This often prevents the link with other sea level data sources (in-situ and satellite data). Previous studies have examined the HDM bias using different methods such as interpolation and deep learning (Jahanmard

et al., 2021; 2022; 2023a).

A sea level correction for Nemo-Nordic model was obtained using the deep learning method in Jahanmard et al. (2023a), which identified a constant vertical reference bias of 18.1 cm and a correction for hourly modelling errors with a range of about 80 cm. The model correction reduced low-frequency errors relative to observations and minimized the noise in high-frequencies. Overall, the method improved the RMSE of the corrected HDM from 7.6 cm to 3.5 cm relative to tide gauge

records, and from 6.5 cm to 4.1 cm relative to satellite altimetry as an external validation source. These external validations were important for acceptance of the corrected DT model. In this study, we also demonstrate that while correcting the original HDM, we preserved the general dynamics and improved the overall results. Section 2 shows this by (i) validating the geostrophic currents between the original and corrected models, and (ii) illustrating differences in DT variations in selected sub-basins. The corrected HDM, now referred to the geoid, provides better spatial and temporal data compared to

tide gauge and satellite altimetry. This enables the calculation of dynamic water volume (using DT), which is later used in the calculations of the water budget variations and barotropic flows in the following sections.

Inherently for this study, utilization of DT allows us to: (i) examine the dynamic water volume distribution and its seasonal and sub-basin variations; (ii) quantify barotropic inflow and outflow through the Danish Strait; and (iii) estimate river runoff by using the Baltic Sea water balance computation (Omstedt et al. 2004; Reckermann et al., 2011).

The use of DT, referenced to a specific geoid model and origin zero level, allows us to modify the quadratic friction law used for Baltic water exchange at the Danish Straits (Omstedt, 1987; Mohrholz, 2018) by removing the bias term and considering water level of the entire Baltic Sea. This, in essence, allows us to calculate all the barotropic water exchanges. To confirm the characteristics of inflows that occur we also examine in-situ bottom salinity data for the Bornholm Basin and also that of hydrodynamic model data. Previous studies used the difference between Landsort tide gauge station, which roughly





represents the Baltic Sea's mean sea level, and the Kattegat sea level. This approach, without considering a common reference surface for sea level determination and relying on a single-point observation, may introduce errors, offsets, or discrepancies. In this study, we show that accurately quantifying DT relative to the geoid for the entire Baltic Sea (Jahanmard et al., 2023a) enables not only the determination of dynamic volume but also barotropic Baltic water exchange. Moreover, river runoff can be estimated through water balance calculations.

To understand the study area a bit more recall that the Baltic Sea is a highly stratified estuary, and despite the limitation in vertical mixing, water still recirculates by means of the 'Baltic Sea Haline conveyor belt' where saline incoming water propagates through the Sounds (Oresund, Great Belt, Little Belt) that upwells in the Baltic and returns out through the straits as brackish water (Döös et al., 2004). Under favourable wind condition Major Baltic Inflow can occur in the deep layer waters of the central Baltic Sea. As a result, both barotropic and baroclinically driven inflows can transport saline water into

the halocline or below it, which depends on the density of the inflow water (Reissmann et al., 2009). The inflows of saline water are forced by winds from the west and outflows are by winds from the east.

The winds, when strong enough, can actually reverse the Ekman transport. For example, persistent westerly winds of 2-5m/s can stop the almost constant surface outflow layer of brackish water (Lehmann et al., 2012). Also, due to the freshwater surplus, the water volume of the Baltic Sea will increase even though no direct inflow takes place, and this will affect all

sub-basins (will be discussed in Figure 4). As a result, atmospheric forcing plays a major role in the dynamics of the Baltic Sea, it more or less depends on the exact location of the polar front and the strength of the westerlies. Thus, the NAO index and the Baltic Sea Index can to some extent characterize the variations observed in the Baltic Sea (Lehmann et al., 2002; Leppäranta and Myrberg, 2009).

This study aims to use DT of the corrected HDM for 2017-2021.5 to quantify: (i) the dynamic water volume of the Baltic

Sea; (ii) the seasonal and spatial distribution of DT anomalies; (iii) barotropic water exchange between the Baltic and North Seas; and (iv) total river runoff to the Baltic Sea using the water budget equation. Additionally, examining the seasonal balance of water budget components can reveal biases in existing models. The present paper first introduces the method with some relevant background concepts described in section 2. This is followed by the results in section 3 and a conclusion in section 4.

## 2 Method

### 2.1 Dynamic Topography Background and Verification for Baltic Sea

The geoid is a shape of the equipotential ocean surface under the influence of gravity and rotation of Earth alone. Therefore, its interpretation represents the natural zero vertical datum for sea level. This implies that any sea level deviation from the geoid (e.g., due to winds, tides, river discharge) is expressed by dynamic topography (DT).

Although the geoid is what realistically should be used for expressing physically meaningful heights and depths, in reality different sea level sources (e.g., tide gauges, HDMs, and satellite altimetry) refer to various vertical datums. For instance,



tide gauge records are often referenced to national chart datums, which may use the mean sea surface, the geoid, or the lowest astronomical tide as the zero-level surface. Satellite altimetry derived sea surface height is usually referred to some reference ellipsoid, and a geoid model is required to compute DT (Mostafavi et al., 2023). HDMs tend to ideally present sea

level with respect to a constant geopotential $W_{as}$ its implicit vertical reference surface (Hughes and Bingham, 2008). Hence, the sea level derived from HDMs is often referred to as DT. However, the vertical reference surface of an HDM may differ from that of a geoid model in its origin, which was quantified by RefBias in Jahanmard et al. (2023a). In addition, the HDMs may be subject to modelling errors due to numerical modelling limitations when compared to in-situ and satellite altimetry.

Accordingly, employing a consistent geopotential reference between diverse sea level data sources facilitates the direct

fusion of simulated and observational data. This approach not only enables a more accurate and realistic determination of DT but also ensures consistency and accuracy in marine areas. This is especially necessary for countries sharing common marine areas, such as the Baltic Sea (Jahanmard et al., 2021). For more information on the vertical reference datum and sources of sea level data, readers are guided to the references highlighted in text above.

In this study, we used Nemo-Nordic model (NEMO NS01) obtained from Swedish meteorological and hydrological institute

(SMHI) and corrected hydrodynamic model from Jahanmard et al. (2023a). The corrected model was adjusted using a deep learning model to reduce the modelling errors with respect to geoid-referenced tide gauge observations (dataset available: Jahanmard et al., 2023b). The deep learning model resolved the errors (with a range of about 80 cm) through finding relationships between spatiotemporal variables, such as winds and sea level pressure, and modelling errors observed at location of tide gauges.

The assessment of the corrected model demonstrates notable spatial and temporal improvements with respect to satellite observations with a root mean squared errors of 4 cm, which limited to less than daily errors. In addition, the corrected model presents an average correlation of 0.98 with 52 tide gauge records, while the original hydrodynamic model shows 0.93 (for more details refer to Jahanmard et al. 2023a). We will demonstrate the calculations of the geostrophic surface currents as an additional validation for the corrected model. This was done to verify the corrected model's adherence to the

quasi-steady circulation patterns observed in the Baltic Sea. Surface currents in the Baltic Sea are influenced by sea surface tilt, wind stress at the sea surface and thermohaline horizontal gradient of density steered by Coriolis acceleration, topography, and friction (Leppäranta and Myrberg, 2009; Soomere et al., 2011).

In general, the observed mean circulation of the Baltic Sea shows a quasi-permanent cyclonic system in the upper layers, whilst in the lower layer the flow is steered by channels and sills. As a result, the different sub-basins demonstrate different

characteristics that are influenced by the bathymetry and geometry. Different forcing factors (e.g., wind, temperature gradients, etc.) affect the surface layer depending on the time scale considered. For longer time scales (months to years), the baroclinic circulation is largely independent of short-term variation in wind. The long-term mean currents are usually weak with average speed of 5 cm/s. For shorter time scales (1-10 days) the currents react to that of the wind stress. The short-term current velocity also includes seiches and tides with periods ranging from 11 to 31 hours (Liblik et al., 2022; Jönsson et al.,

2008). In addition, the upper layer is characterized as being more under the influences of atmospheric forces. This indicates



that Ekman transport and coastal upwellings/downwellings are also prominent (Döös et al., 2004; Delpeche-Ellmann et al., 2017 and 2021).

The geostrophic current, which represents the balance between pressure gradients and the Coriolis force, can be deduced from the following determination:

$$u_g = -\frac{1}{\rho f}\frac{\partial P}{\partial y} = -\frac{g}{f}\frac{\partial (MDT)}{\partial y}$$
$$v_g = \frac{1}{\rho f}\frac{\partial P}{\partial x} = \frac{g}{f}\frac{\partial (MDT)}{\partial x}$$

(1)

where $\partial/\partial x$ and $\partial/\partial y$ denote the horizontal gradients along the zonal and meridional directions, respectively, in the Cartesian coordinate; and $\rho$, $g$ and $f$ are the mass density, gravitational acceleration and the Coriolis parameter, respectively. The partial derivatives of the $P$ are horizontal pressure gradient, which can be obtained by computing mean dynamic topography ($MDT$) over a given period. The accuracy of geostrophic current computations depends on the precise referencing of the surface topography to a geopotential surface.

In this study, the seasonal geostrophic currents of the Baltic Sea have been calculated using 7-point stencils (Arbic et al., 2012) from seasonal MDT computed from original and corrected Nemo-Nordic model. The pattern of geostrophic current agrees well with the Baltic circulation from other studies (Döös et al., 2004; Soomere and Quak, 2013; Placke et al., 2018; Hinrichsen et al., 2018; Barzandeh et al., 2024). The cyclonic circulation in the Baltic Proper shows the northward currents in the east and southward currents in the west (Jędrasik and Kowalewski; 2019; Liblik et al., 2022). In the Gulf of Finland, a
persistent westward current along the Finnish coast and a narrow eastward current along the Estonian coast can be observed (Alenius et al., 1998; Soomere et al., 2011). Additionally, cyclonic currents are present in the Bothnian Sea, particularly strengthening during the autumn and winter months.

Figure 2 shows the current velocity contours for the corrected model in the top row panels, alongside blue isolines from the original model with a 1 m/s step size. The inset panels show the differences between the models, indicating minimal
deviations limited to specific areas, with a standard deviation of about 0.5 cm/s. Notable differences are observed along the eastern coastline of the Baltic Proper during summer (JJA) and winter (DJF), where the original model overestimates the magnitude of the currents compared to the corrected model. Figure 2 also shows the current direction in the bottom row panels. No significant differences in the direction of the currents were observed between the original and corrected models.



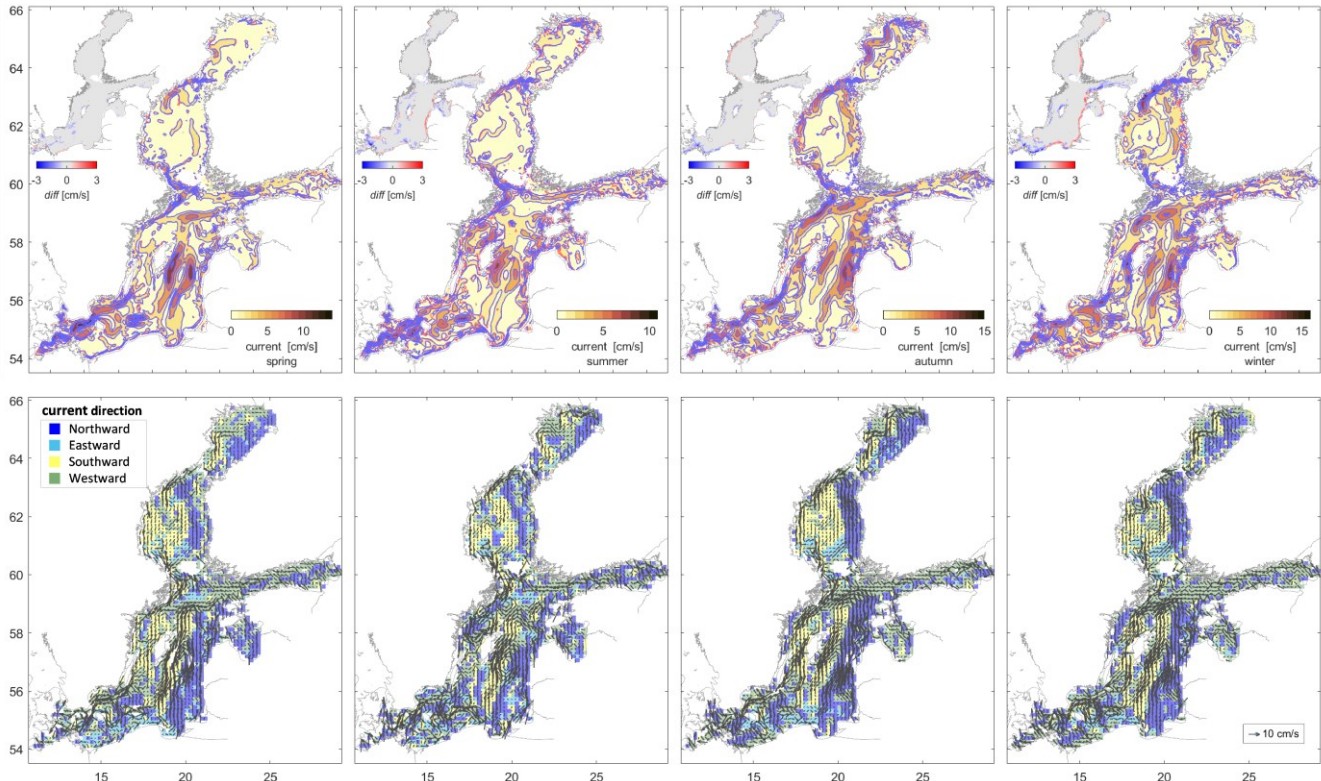

**Figure 2: Seasonal geostrophic currents from 2017.0 to 2021.5. The top row panels show the magnitude of geostrophic velocity from the Nemo-Nordic model, with the corrected model represented by red isolines and colour contour maps, and the original model by blue isolines. The inset panels (upper left of the top row) display the difference between the original and corrected models (original -corrected). The bottom row panels illustrate the direction of geostrophic currents from the corrected model, which is nearly identical to the original.**

As a result, Figure 2 demonstrates that the corrected DT not only improved the accuracy of the simulated sea surface (when compared to satellite altimetry and tide gauge data in Jahanmard et al., 2023a) but also, by correcting the errors and biases of the original DT, we have managed to preserve the underlying circulation patterns.

**2.2 Dynamic Water Volume Background**

The Baltic Sea has an average volume of 21,205 km$^3$, with an annual addition of about 480 km$^3$, mainly from river runoff and net atmospheric flux (precipitation minus evaporation). The volume decreases via outflows through the Danish Straits. The total freshwater budget remains consistently positive due to substantial river runoff, with monthly runoff ranging from 0.85 to 2.16 km$^3$/day (Leppäranta and Myrberg, 2009).

Conventionally, the water volume is computed by integrating the water column from the seafloor to the sea surface. To study the water balance of a basin, it is beneficial to separate the total water volume into two components: the dynamic water volume $V(t)$, which fluctuates over time, and the constant water volume $V_0$. These two components can be distinguished



using an equipotential surface (e.g., a geoid model). Therefore, considering DT as a sea level variation relative to a stable geoid-based vertical reference surface, the dynamic volume is calculated through the spatial integration of DT fluctuations. Hence, the constant water volume is the integration of water columns from seafloor to the geoid surface.

In fact, the $V(t)$ represents the volume variations from $V_0$ due to factors such as inflow/outflow, river runoff, and net atmospheric flux. Note the constant volume can also experience changes over the years due to sediment redistribution and/or vertical land movements (e.g. glacial isostatic adjustment). However, its dynamic effects on the basin are reflected in the dynamic component, as the geoid surface is (quasi)static. The slow changes in the geoid over time due to factors, such as glacial isostatic adjustment and variations in Earth's mass distribution, can be accounted for in the computations (Kakkuri and Poutanen, 1997). Therefore, total water volume $V(t)$ is determined as follows:

$$\boldsymbol{V}(t) = V_0 + V(t) = \iint\limits_A H(x,y) \; dxdy + \iint\limits_A DT(x,y,t) \; dxdy \tag{2}$$

where $H$ represents the charted depth relative to the geoid surface, and $x$ and $y$ represent the Cartesian zonal and meridional coordinates, respectively. In this study, the focus is on the utilization of dynamic water volume $V(t)$, for before it was not accurately defined with respect to the geoid.

To simply demonstrate the difference in volume calculations between original Nemo-Nordic model and corrected model, Figure 3 compares the dynamic water volume of these data. To align origin zero levels between the original Nemo-Nordic model and the corrected model, the reference bias (18.1 cm) was removed from the original model (see Jahanmard et al., 2023a for details). Figure 3 shows that the dynamic water volume of the Baltic Sea varies from -75 km$^3$ to 340 km$^3$. The corrected DT, validated by satellite altimetry measurements, reveals a seasonal error in water volume estimation by the Nemo-Nordic model, ranging from -60 km$^3$ to 50 km$^3$.



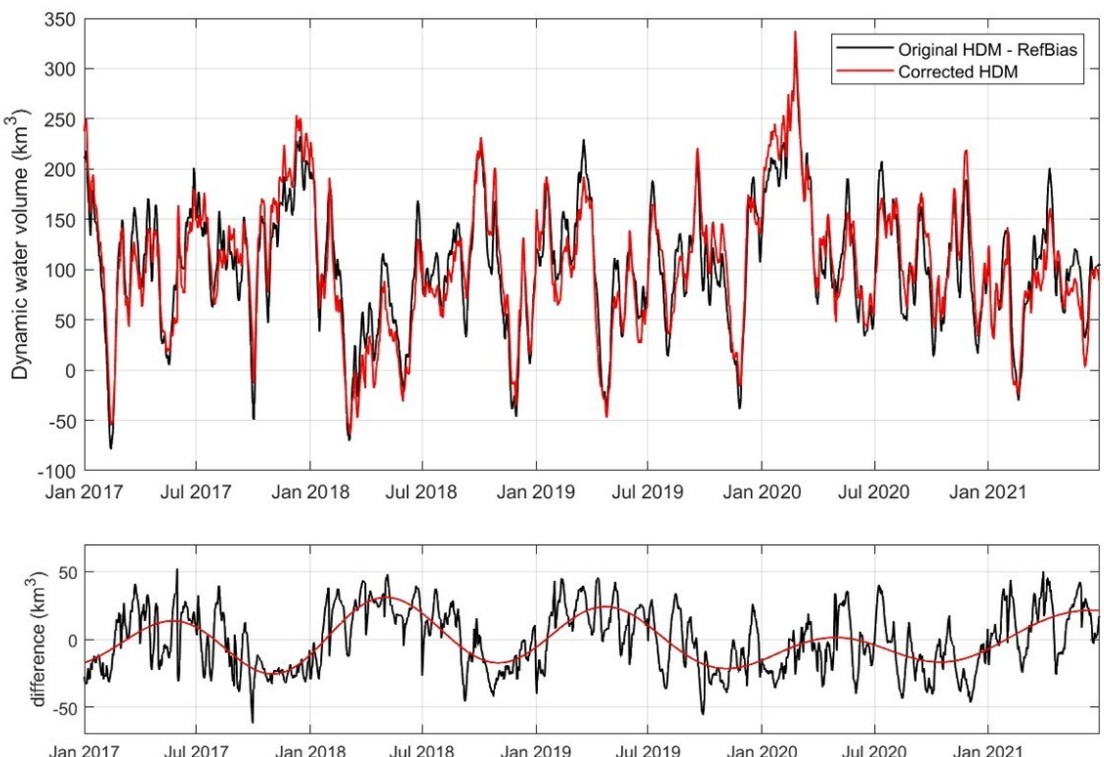

**Figure 3: Dynamic water volume $V(t)$ of the Baltic Sea during the designated period. The top panel shows the water volume from the Nemo-Nordic model (black) and the corrected model (red). The bottom panel presents the discrepancy between the two models (original – corrected) in black, whereas its smoothed version is in red.**

Figure 3 (bottom) shows that the original Nemo-Nordic model tends to overestimate water volume during spring and summer, and underestimate it during autumn and winter months. A change of 10 km$^3$ in water volume is associated with an approximate sea level variation of 2.6 cm based on geometry of the study area.

This seasonal trend of over- or underestimation in the original Nemo-Nordic model hints specific drivers. For instance, peak river discharge in the Baltic Sea, primarily driven by atmospheric precipitation and snowmelt, occurs between April and June (Graham, 2004; Raudsepp et al., 2023). This may have introduced a seasonal bias in the model. Additionally, the discrepancy could be related to underestimating wind forces, as south-westerly and westerly winds dominate in autumn and winter, usually resulting in sea level accumulation in the eastern and northern regions of the Baltic Sea (Alenius et al., 1998). Further investigation is needed to confirm these hypotheses.

To examine variation in the dynamic water volume for each sub-basin, their volume was normalized by the sub-basin area $A_b$. This term is equivalent to the spatial mean of DT for each sub-basin ($DT_b$), with the distinction that meridian convergence is also considered. Therefore:

$$DT_b(t) = \frac{1}{A_b} V_b(t) \tag{3}$$





where $V_b(t)$ is the dynamic water volume obtained from Eq. (2) for sub-basin $b$. $DT_b$ enables the determination of the water balance between sub-basins, as discussed in Section 3.1. Notably, using the geoid surface as a reference provides stable, physically meaningful measurements of dynamic variations between sub-basins due to factors such as winds, density gradients, and currents.

Figure 4 shows the seasonal variation of $DT_b$ from the Nemo-Nordic model (corrected) and the average tide gauge readings

from sub-basins. Key findings include: (i) the corrected DT closely follows the seasonal pattern observed by tide gauges; (ii) sub-basin DT is higher in autumn and winter; and (iii) water levels in the northern sub-basins (Bothnia Bay/Sea and Gulf of Finland) are consistently higher than in the southern sub-basins (Bornholm Basin and southern Baltic Sea). This is mainly due to permanent horizontal water density differences, driven primarily by salinity, which result in a higher sea level in the north. On average, sea level is expected to decline by 35-40 cm from the Bay of Bothnia to the Skagerrak (Leppäranta and

Myrberg, 2009), which is partially reflected in Figure 4, where the difference between Bothnia Bay and Bornholm Basin is 24 cm in winter and 13 cm in spring. Additionally, the original Nemo-Nordic model displays different seasonal variations compared to the tide gauges and the corrected model, with its discrepancy remaining relatively constant across sub-basins. The difference between the tide gauge and the corrected model arises because the corrected model averages DT across the entire sub-basin, whereas tide gauges reflect DT variation at specific locations. These two measures align when the corrected

model is averaged at the tide gauge locations.

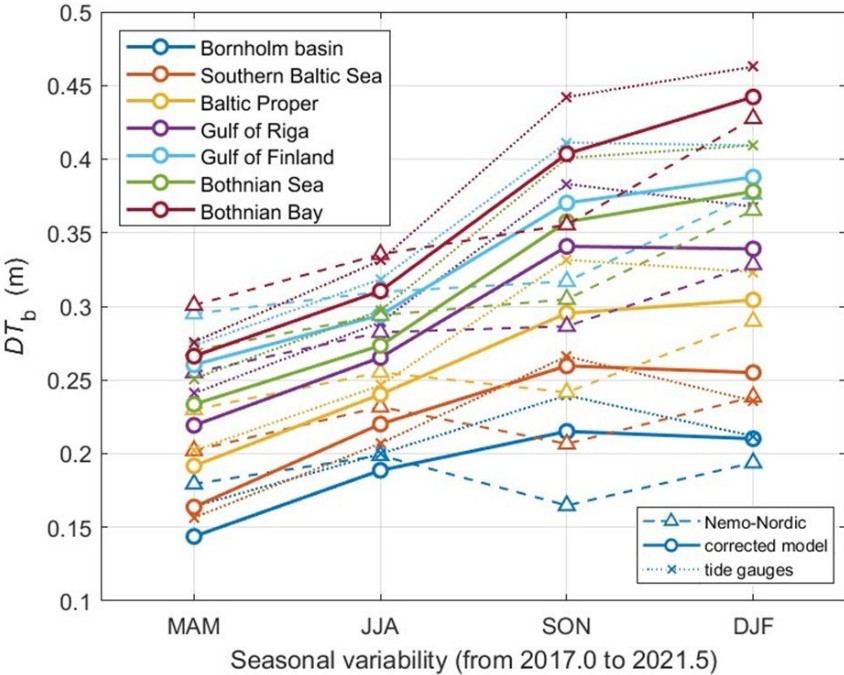

**Figure 4: Seasonal $DT_b$ computed from Eq. (3) for the Baltic sub-basins, represented by different colours. Dashed, solid, and dotted lines denote DT values from the original Nemo-Nordic model, the corrected model, and tide gauge observations, respectively.**





Figure 4 also shows that Baltic Sea water level increases in autumn (SON) and winter (DJF) and decreases in spring (MAM) and summer (JJA). This seasonal increase in autumn and winter months is attributed to: (i) dominant south-westerly and westerly winds piling up water in the eastern and northern sub-basins; (ii) river inflow is still significant, and the freshwater budget remains positive (Leppäranta and Myrberg, 2009); and (iii) Baltic inflow has the trends of being highest for these months. The following section will explore the transport of water volume between the Baltic Sea and the North Sea in more
detail.

One can compute the spatial anomaly of $DT_b$ by subtracting the spatial mean DT of the entire Baltic Sea. The variable DT anomaly ($DTa$) represents relative variations in DT among the sub-basins. This can indicate co-oscillations between sub-basins and assess the correlation between DT distribution in the Baltic Sea and factors such as wind. Therefore:

$$DTa_b(t) = DT_b(t) - DT_{BS}(t) \qquad\qquad (4)$$

where $DT_{BS}$ represents the spatial mean DT (according to Eq. 3) over the entire Baltic Sea. The variable of $DTa_b$ can in fact
serve to illustrate the internal dynamics of the Baltic Sea and reveals the co-oscillations among its sub-basins.

**2.3 Water Exchange between Baltic Sea and North Sea**

The saline water exchange between the Baltic Sea and North Sea typically occurs between the Kattegat and the Danish Straits. The exact exchange location varies, for the exchange can occur in the Kattegat-Danish Straits or it may also be considered at the Danish Strait and the Baltic Sea. This study examines the barotropic water exchange between the Baltic
and North Seas, where the Baltic Sea maintains equilibrium with the open ocean through the shallow, narrow Danish Straits (Omstedt et al., 2014). These straits act as a low-pass filter for the Baltic Sea, preventing the entry of high-frequency variations from the Nordic Sea (Weisse & Hünicke, 2019). Water exchange is driven by differences in sea level (barotropic) and density gradients (baroclinic) between the Kattegat and Arkona Basin. Baroclinic events, primarily driven by salinity gradients, occur mainly during calm summer conditions. For most of the year, barotropic forcing exceeds baroclinic forcing
considerably, as wind forcing (especially zonal) and air pressure establish sea level differences between the Kattegat and Baltic Sea (Mohrholz 2018; Leppäranta and Myrberg, 2009).

Due to the anisotropic wind properties of the Baltic Sea (Soomere, 2003) and the requirement for favourable conditions that allow water exchange between the Baltic and North Sea (Zhurbas and Väli, 2022), these input/output events typically occur intermittently on different scales (small, medium, or large). Also, inflows have a seasonal trend, with intense events usually
in November-January and a minimum in May. Inter-annual/intra-annual variations also occur, which are influenced by air pressure, winds, sea-level differences between the Baltic and North Sea.

However, replenishing the deep bottom waters of the Baltic Sea requires a large inflow, often referred to as a Major Baltic Inflow (MBI). Such MBIs typically occur once per decades, and apart from increasing the total water volume in the Baltic Sea, they also bring salty, oxygen-rich waters and dense water to the deep areas of the Baltic, extending as far as the Gotland
Basin (Purkiani et al., 2024). There was no MBI during the period of this study, therefore, our focus is not on determining MBI, for which several methods have been employed (Matthäus, 1993; Lehmann and Post, 2015). Instead, we demonstrate



how utilizing corrected HDM model can be used to quantify all the barotropic water exchanges between the Baltic Sea and North Sea basins. Determination of DT, which refers to an identical geoid reference surface, allows us to accurately calculate sea level slopes between the two basins.

Quantifying volume transport through the Danish straits requires consistent time series of the mean sea levels of the Baltic Sea and Kattegat, along with river runoff data (Mattsson, 1996; Mohrholz, 2018). In Mohrholz (2018), the mean sea level time series for the Baltic was derived from tide gauge measurements at Landsort, which is a reasonable representation of the mean sea level of the Baltic. This study employs a similar approach with some modifications. The main difference is the use of the mean DT of the entire Baltic Sea from the corrected model, and the concept of using a stable geoid-based reference

surface, which allows an accurate determination of dynamic water volume and DT inclination between the Baltic Sea and Kattegat basin. Thus, making the quantification simpler and more accurate. As a result, the barotropic flow through the Danish Straits can be quantified by the quadratic frictional law:

$$Q(t) = \sqrt{\frac{DT_{BS}(t) - DT_K(t)}{K_f}} \tag{5}$$

where $Q$ represents the flow rate, and positive values indicate outflow from the Baltic Sea. $DT_{BS}$ and $DT_K$ are respectively DT of the Baltic Sea and Kattegat, and $K_f$ is the empirical flow resistance coefficient (Stigebrandt, 1983). In this equation,

offset correction is no longer needed, as both DT measurements were taken on a common reference surface. The value of $K_f$ is $2.03 \times 10^{-10}$ s2m$^{-5}$ with the uncertainty of approximately 10% (Mattsson, 1995 and 1996). Hourly time series of $DT_{BS}$ is computed by Eq. (3) for the entire Baltic Sea, and $DT_K$ is determined by averaging geoid-referenced tide gauge readings located in Kattegat (see Figure 1).

**2.4 Water Budget of the Baltic Sea**

The water budget of the Baltic Sea consists of several main parameters such as: river runoff $R$, evaporation $E$, precipitation $P$, and inflow/outflow through the Danish Straits $Q$ (see Eq. 6). The two main parameters are that of Baltic inflows/outflows and mean river discharge. In fact, the mean annual river discharge of 436 km$^3$ is almost as dominant as the total inflow of saline water from the North Sea. The net atmospheric flux (precipitation and evaporation) is about 10 times smaller than the river inflow, and it is positive values from January to August and negative from September to December (Leppäranta and

Myrberg, 2009). As a result, the total freshwater budget of the Baltic Sea, primarily dominated by river inflow, consistently remains positive on monthly timescale. The following simplified equilibrium equation can express the conservation of Baltic water mass and account for temporal variations in water volume (Omstedt et al, 2004; Reckermann et al., 2011; Mohrholz, 2018):

$$\frac{dV(t)}{dt} = R(t) + (P(t) + E(t)) - Q(t) \tag{6}$$



where $R$, $P$, $E$, and $Q$ are the rates of river runoff, precipitation, evaporation, and Baltic flow rate, respectively. Given the
water volume time series $V(t)$ was derived from Eq. (2) and Baltic flow rate $Q(t)$ determined by Eq. (5), the rate of total
river runoff can also be derived from accurate DT of the Baltic Sea, as follows:

$$R(t) = \frac{dV(t)}{dt} - \big(P(t) + E(t)\big) + Q(t) \qquad (7)$$

The net atmospheric flux $(P + E)$ is the sum of the precipitation and evaporation, where the positive evaporation is
downward. The precipitation and evaporation datasets were downloaded from ERA5 hourly data (Hersbach et al., 2023). The
precipitation parameter is the accumulated rain and snow over the Baltic Sea. The precipitation and ice melting on land, as
well as groundwater flow, are included in the total river runoff. Comparing Baltic in/outflows and total river runoff,
computed from the original and corrected Nemo-Nordic models, can also provide insights into the source of seasonal bias in
sea level modelling (see Figure 3 and 4), as will be shown in the Results section.

## 3 Results

The utilization of accurate DT that are referred to the geoid, is expected to allow a more accurate quantification and
examination of the Baltic sea water budget components. Such a method, to our knowledge (Section 2) hasn't before been
utilized. In this section, we focus on: (i) examination of the spatial DT anomaly for different sub-basins of the Baltic Sea; (ii)
quantification of all barotropic exchanges between the Baltic Sea and North Sea; (iii) utilization of water budget equation to
derive river runoff; and (vi) examination of seasonal distribution of the dynamic water volume.

### 3.1 Dynamic Topography Anomaly of Sub-Basins

Dynamic topography variations of the Baltic Sea are governed by several factors functioning on different timescales, with
both periodical and irregular frequencies. The sub-basins also display spatial variations in DT, influenced by permanent
features such as geometry, bathymetry, and location. Recall that the Baltic Sea typically has a persistent northward (and
eastward) DT tilt (Figure 4), primarily induced by halosteric effects due to voluminous freshwater river discharge into the
northern (and eastern) regions. Alongside this, atmospheric factors such as air pressure, winds, and variations in salinity and
temperature significantly influence DT variation from hourly to seasonal timescales. Hence, examining the seasonal
distribution of DT variations across the different sub-basins is important for a better understanding of the underlying
physical patterns. Table 1 presents the mean DT of the Baltic sub-basins, and their seasonal and annual distribution obtained
by applying Eq. (4).

**Table 1: Annual and seasonal average dynamic topography (in meters) for sub-basins. Numbers in parentheses indicate the
difference between seasonal and annual means. The sub-basins are listed from south to north (see Figure 1).**

| Sub-basin (Avg. depth) | Annual mean | Spring MAM | Summer JJA | Autumn SON | Winter DJF |
|---|---|---|---|---|---|
| Bornholm basin (32) | 0.19 | 0.14 (-0.05) | 0.19 (0.00) | 0.22 (0.03) | 0.21 (0.02) |





| | | | | | |
|---|---|---|---|---|---|
| Southern Baltic Sea (56) | 0.23 | 0.16 (-0.07) | 0.22 (-0.01) | 0.26 (0.03) | 0.26 (0.03) |
| Baltic Proper (78) | 0.26 | 0.19 (-0.07) | 0.24 (-0.02) | 0.30 (0.04) | 0.30 (0.04) |
| Gulf of Riga (24) | 0.30 | 0.22 (-0.08) | 0.27 (-0.03) | 0.34 (0.04) | 0.34 (0.04) |
| Gulf of Finland (37) | 0.34 | 0.26 (-0.08) | 0.29 (-0.05) | 0.37 (0.03) | 0.39 (0.05) |
| Bothnian Sea (61) | 0.32 | 0.23 (-0.09) | 0.27 (-0.05) | 0.36 (0.04) | 0.38 (0.06) |
| Bothnian Bay (44) | 0.37 | 0.27 (-0.10) | 0.31 (-0.06) | 0.40 (0.03) | 0.44 (0.07) |

Spatial anomalies in DT ($DTa$ from Eq. 4) allow for the analysis of oscillations amongst sub-basins at different timescales. This is done by subtracting the spatial mean DT of the entire Baltic Sea from the DT of any given sub-basin at an identified

timescale. As a result, $DTa(t)$ represents the water level of sub-basins in relation to each other. Figure 5 illustrates monthly DTa of the sub-basins shown in Figure 1. Observe a consistently positive DT in the northern and eastern sub-basins, and negative values in the southern region of the Baltic Sea, along with the co-oscillation between sub-basins on monthly timescale. To emphasize the greatest difference observed between basins we highlight an extreme case that occurred in February 2020, where the difference between the DT in the northern and southern parts of the Baltic Sea reached up to 50

cm. Figure 5b shows the seasonal perspective, where an obvious increase in sea surface tilt during autumn and winter months occur, with a maximum difference of 32 cm between the northern and southern basins in December. The occurrence of these maximum difference usually takes place during the autumn and winter months, when predominant south-westerly and westerly winds are the strongest (Leppäranta and Myrberg, 2009). Contrary, the sea surface shows minimal $DTa$ inclination in summer months. In general, the Bornholm and Bothnian Bay basins experience the highest variability

throughout the year (32 and 29 cm, respectively), while the Baltic Proper shows the least variation.

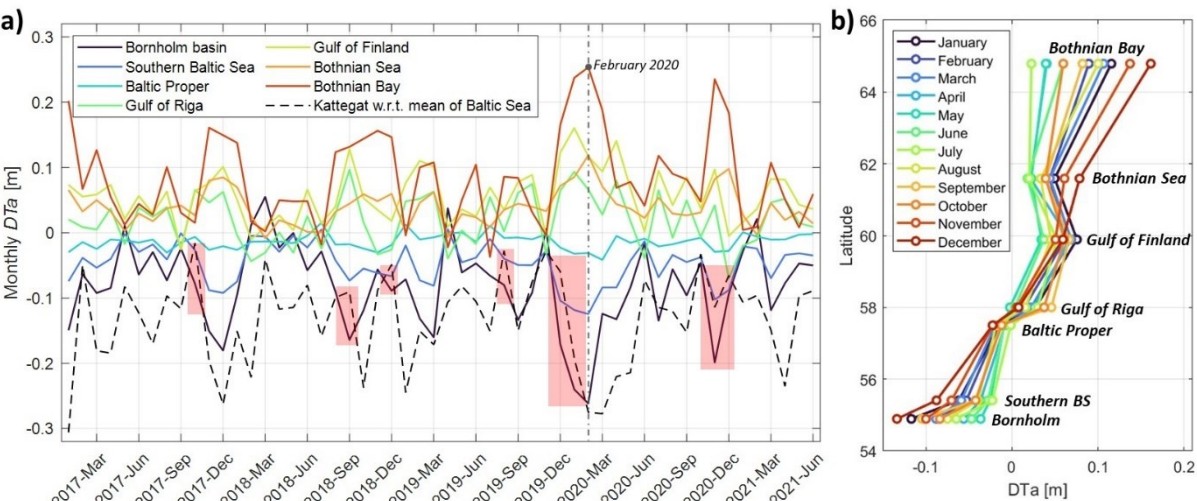

**Figure 5: Spatial anomalies in DT of sub-basins. a) monthly mean of $DTa$ for different sub-basins, along with mean DT of Kattegat with respect to the mean of Baltic Sea. b) Climatological monthly mean $DTa$ for the designated time period plotted against the latitude midpoint of sub-basins.**



The Kattegat and Danish Strait are known as transition areas that allows the exchange of the salty North Sea waters and fresher Baltic Sea waters. The variation in DT of Kattegat relative to the mean of the Baltic Sea is also depicted in Figure 5a by a dashed line. Observe that this variation consistently remains below the average water level of the Baltic Sea. This positive sea level compared to the Kattegat, leads to a predominant outflow from the Baltic Sea. However, during some events on this monthly timescale, the $DTa$ of Kattegat surpasses the southern sub-basins. This consequently causes the

inflow of saltwater into the southern Baltic basin (highlighted in Figure 5a by red boxes). These results of the exchange of the Baltic and North Sea waters are examined deeper in in section 3.2.

From a spatial context, Figures 6 shows the average $DTa$ and its standard deviation at model grid points, computed from hourly data. On average, the northern and eastern sections of the Baltic Sea display a consistently positive dynamic topography of about 10 cm relative to the mean states of the Baltic Sea, while the southern part exhibits lower $DTa$ (about -

10 cm) than the Baltic average. Furthermore, Figure 6b illustrates the variability of $DTa$ across the Baltic Sea, which can highlight the first barotropic basin mode (Wubber and Krauss, 1979) under influence of atmospheric forcing. High-frequency DT variations in the Baltic Sea are internally isolated due to the characteristics of the Danish Straits (Weisse & Hünicke, 2019), which result in the Baltic seiches (Jönsson et al., 2008). The variability of $DTa$ increases to over 15 cm by distancing from the Baltic Proper. Greater $DTa$ variability may indicate areas with a higher potential to be impacted by

extreme sea level events. Furthermore, areas with high standard deviation along with the average $DTa$, as shown in Figure 6a and b, coincides with some of the identified areas of coastal erosion especially that in the Gulf of Finland, Gulf of Riga and southern Baltic Sea (Weisse et al., 2021; Pindsoo and Soomere, 2020).

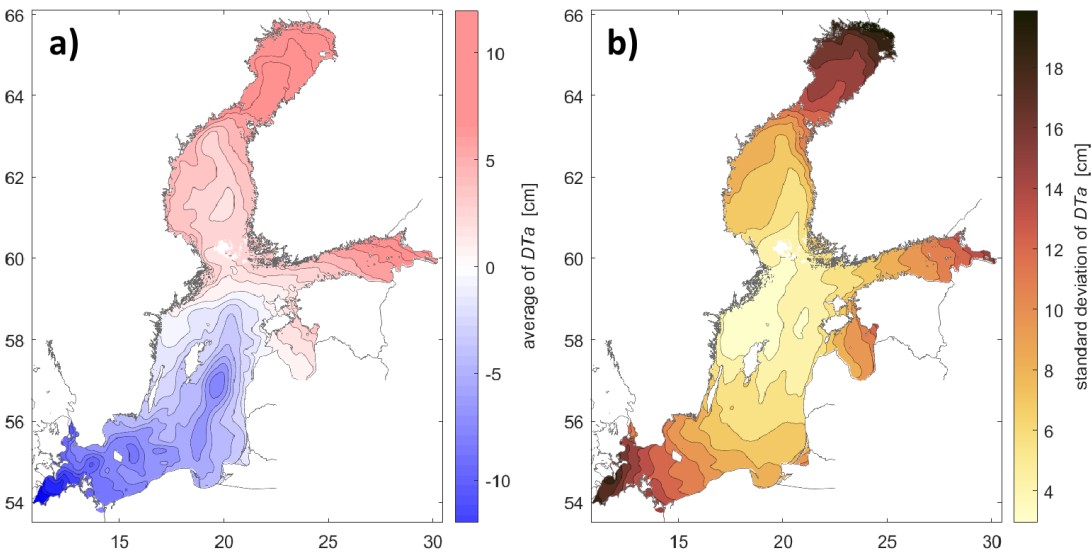

**Figure 6: Temporal mean (a) and standard deviation (b) of $DTa$ (Eq. 4), computed at model grid points based on hourly data.**



Additionally, observed that in Figure 6a for the northern Baltic Proper sub-basin, the $DTa$ values approach around 0 cm and standard deviation of less than 4 cm. This is also complemented by Figure 5b which shows that the seasonal variation in the Baltic Proper is very small compared to other sub-basins. These observations signify that for this particular region in the Baltic Proper can be used to closely represent an equilibrium mean DT for the entire Baltic Sea. This indicates that any significant changes in DT values in this area may be used as climatic changes indicators or for the occurrence of ocean

dynamics of the BS (e.g. Baltic inflows).

The co-oscillation amongst sub-basins is determined by correlation coefficient of $DTa$ on monthly timescale. Figure 7 shows a strong anti-correlation between southern and northern (eastern) sub-basins of about -0.90 (-0.80). Since the $DTa$ of sub-basins was normalized to the mean sea level of Baltic Sea, the oscillation of $DTa$ is mostly controlled by atmospheric forcing across the entire basin. Figure 7b shows the correlation between sub-basin $DTa$ and the predominant wind over the Baltic

Sea. The northern (eastern) regions display a strong correlation of 0.85 with meridional (0.94 with zonal winds). This can indicate that the highest $DTa$ variation and the monthly dynamics of these sub-basins are mostly driven by winds, especially that of westerly and south-westerly winds which pile up water in the northern and eastern sub-basins. Also, strong anti-correlations between $DTa$ of southern basins and winds is observed.

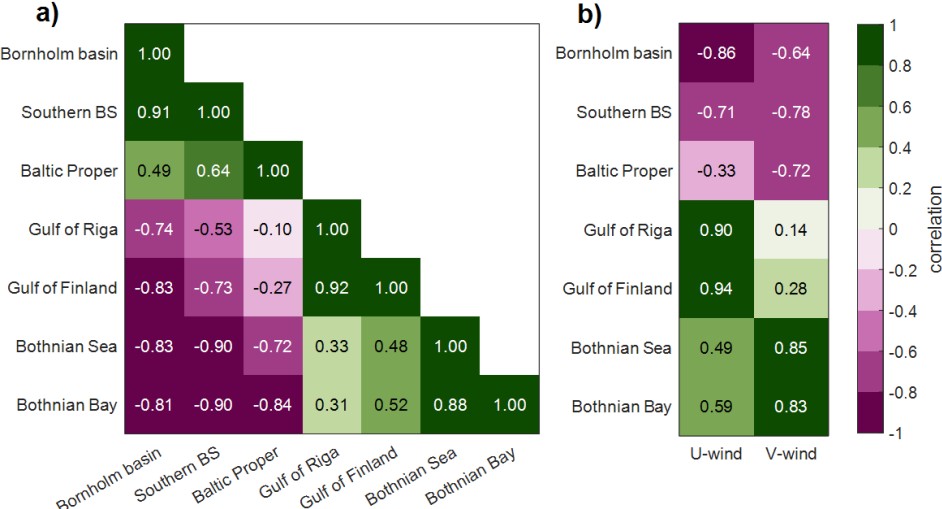

**Figure 7: Correlation of sub-basins' $DTa$ on a monthly timescale (a), and the correlation of $DTa$ with the predominant wind across the Baltic Sea.**

## 3.2 Barotropic Water Exchange Between Baltic and North Sea

As was described in Section 2.3, the DT inclinations between the Kattegat and the Baltic sea can be used to indicate the occurrences of barotropic water exchanges (inflow and outflow) between the Baltic Sea and the North Sea. This can be

accomplished by computing the flow rate ($Q$) based on Eq. (5). Thus, the computed barotropic Baltic flow rate through the



Danish Straits for the period of 2017-2021.5 is presented in Figure 8. Note that for the period examined in this study, no Major Baltic Inflow occurred, thus the results described below relate to medium and small exchanges.

At first glance, the average outflow and inflow rates display high frequency signals (Figure 8a). However, by applying specific temporal criteria to filter out the high-frequency variations, it becomes possible to distinguish the relevant inflow

and outflow events. For the statistical analysis, the inflow events (when negative) and outflow events (when positive) can be distinguished using the criterion outlined in Fischer and Matthäus (1996), where each event should last at least a day (as indicated by the black horizontal lines in Figure 8a). Therefore, the histogram of event durations (Figure 8b) presents the number of inflow and outflow events based on their duration. In Figure 8a, another criterion for classifying each event that lasts at least 5 days is also indicated by the green lines.

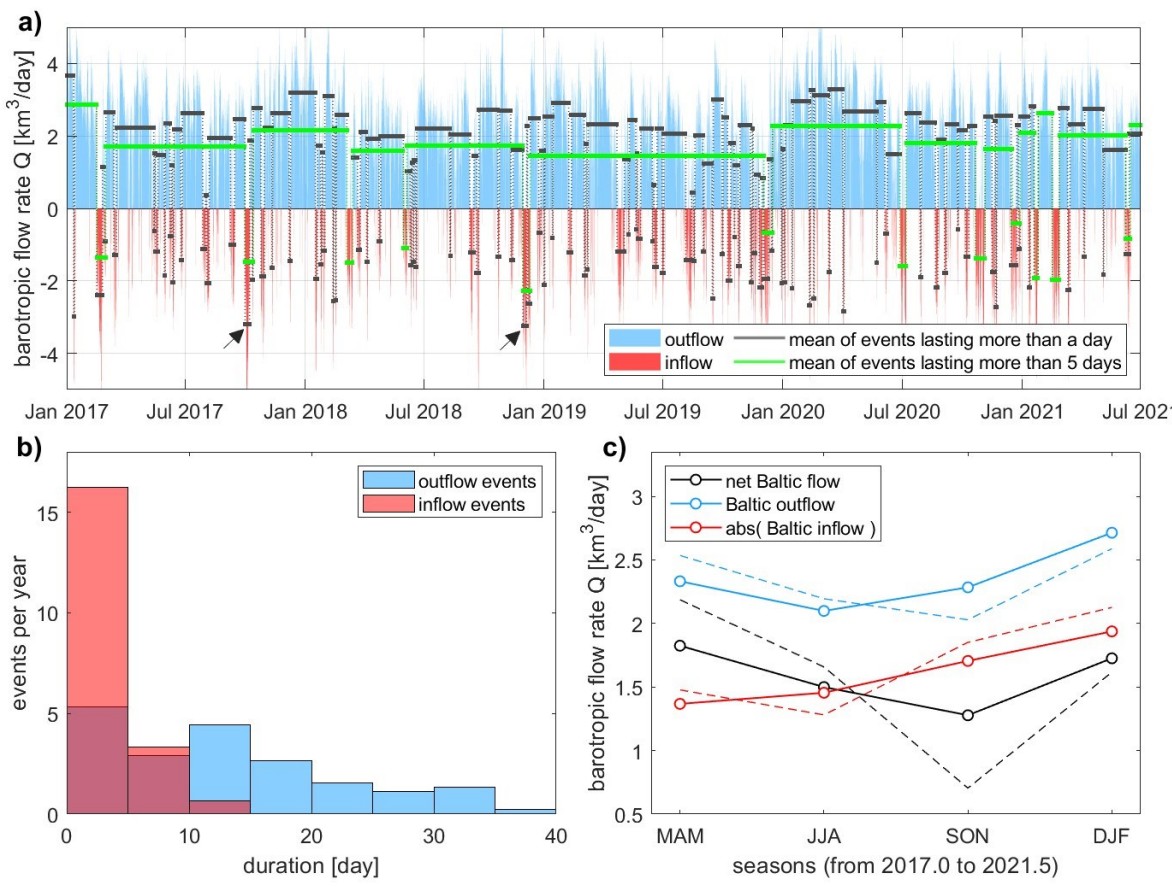


**Figure 8: Barotropic Baltic water exchange calculated from hourly time series of dynamic topography inclinations between the Baltic Sea and the Kattegat basin. a) Time series showing positive (outflow, in blue) and negative (inflow, in red) Baltic flow rates, along with the mean values of events lasting more than one and five days. b) Histogram of event duration per year for events longer than one day. c) Average flow rates over seasonal timescales, with solid lines for the corrected model and dashed lines for**
**the original model.**



Figure 8b shows that most inflow events, based on the one-day lasting criterion, have a duration of less than 5 days, with about 16 events occurring per year. For inflow duration of more than 5 days, the number of occurrences is about four events per year; and less than one for duration lasting more than 10 days. However, this result is sensitive to changes in event classification criteria, which subsequently alter the event histogram. Nevertheless, the overall inflow pattern confirms that

Baltic Sea inflow events are sparse and brief.

Two significant inflow events, in terms of duration and magnitude, occurred in October 2017 and November 2018 (denoted by arrows in Figure 8a). Each event lasted about a week, with inflow rates of 3.19 $km^3$/day and 3.24 $km^3$/day, respectively. The total volume of inflow water for the events was approximately 22.5 $km^3$, which is small compared to Major Baltic Inflows (e.g., 198 $km^3$ in December 2023 for 14 days; Purkiani et al., 2024) but comparable to the average daily outflow

(2.36 $km^3$). Recall that due to the predominantly positive freshwater budget dominated by river runoff, the Baltic Sea consistently experiences outflow. These outflows occur at different rates with a seasonal pattern (Leppäranta, M. and Myrberg, K., 2009).

Figure 8c shows the seasonal mean values of the calculated Baltic flow rate, with an average outflow of 2.36 $km^3$/day and inflow of 1.6 $km^3$/day. These values are closely aligned with the long-term mean water exchange reported by Liljebladh and

Stigebrandt (1996), which are 2.59 $km^3$/day for outflow and 1.3 $km^3$/day for inflow. In winter, both inflow and outflow rates are in maximum value due to increased wind variation and storm conditions. Concerning the net Baltic flow, a consistent positive outflow is observed on a seasonal timescale, with a maximum in spring and a minimum in autumn. Interestingly, the black dashed line in Figure 8c represents the net Baltic flow form the original Nemo-Nordic model, which indicates a significant difference in autumn.

To investigate the occurrence of barotropic inflow events with salinity changes in the southern sub-basins, we also examined the variation of bottom salinity. The salinity data was obtained from the Baltic Sea Physics Reanalysis model (2024) and this model shows a good agreement with the CTD measurements at station BY2 located in Bornholm Basin (as shown in Figure 9a). The spatial basin average of bottom salinity is shown in Figure 9b for the southern sub-basins, along with inflow/outflow rates with lasting of greater than 1 day from Figure 8a in the background. One can observe a pattern in the

salinity anomalies for the bottom layer of the Bornholm basin and the Southern Baltic Sea, which increases with sustained inflow and decreases when there is no significant inflow. This observation in the Southern Baltic Sea basin suggests that the inflow events can reach as far as the southern basins (red line) but are not effective enough to introduce saline water to the deeper waters basin of the Baltic Proper.

A pattern observed is that prior to certain inflow events, such as the one that occurred in October–November 2018

highlighted by the dashed box, the two southern sub-basins are mixing in such a way that the bottom salinity decreases in the Bornholm basin while increasing in the Southern Baltic Sea basin. The rate of bottom salinity change for these two sub-basins are also presented in Figure 9c. This figure shows positive spikes in the Bornholm sub-basin when the 1-day lasting criterion denotes an inflowing event. This pattern, where bottom salinity in these sub-basins starts to mix before an inflow event, may be influenced by factors such as wind and pre-event mixing. Further research can explore this topic.





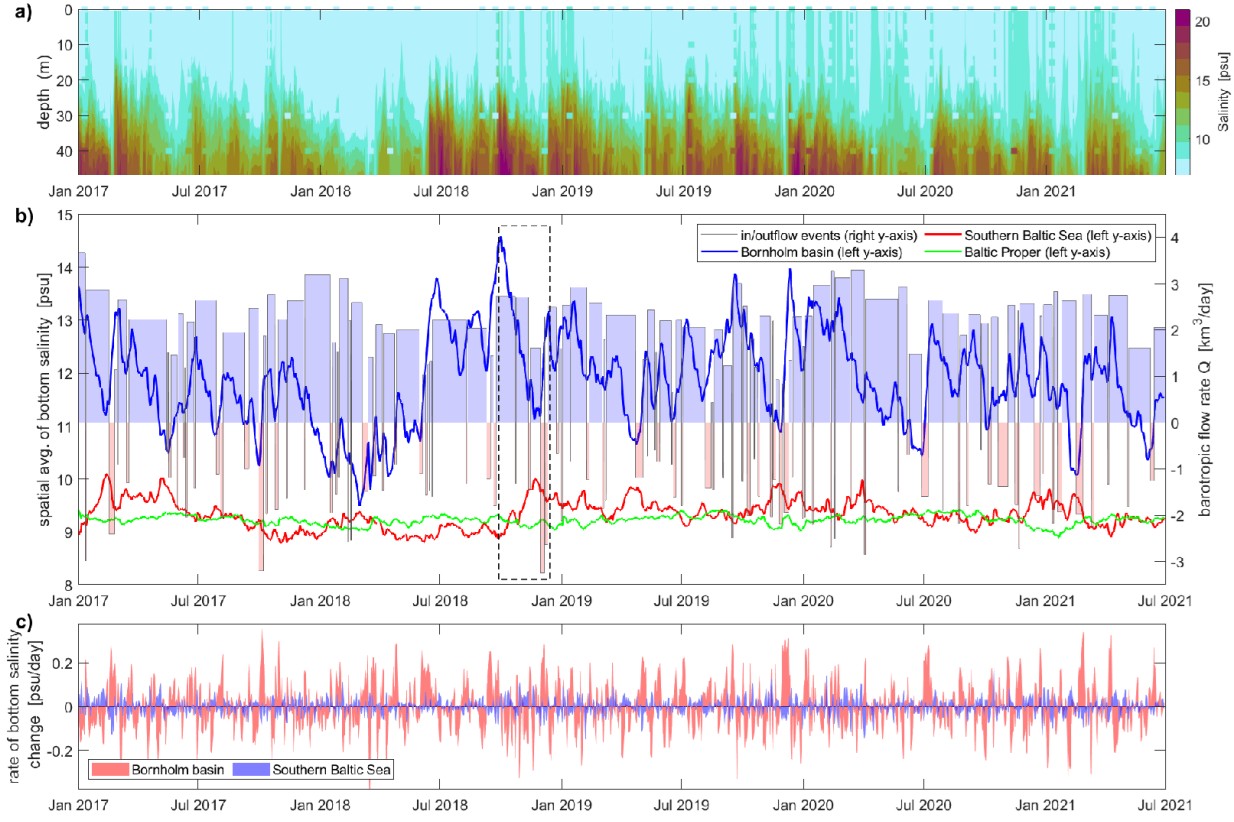

**Figure 9: Bottom salinity: a) Comparison of the Nemo-Nordic model (contour background) and CTD observations (square markers) at BY2. b) Spatial average of bottom salinity in southern basins, along with barotropic flow rate from Figure 8a on the right y-axis. c) Rate of change in (time derivative) bottom salinity.**

### 3.3 River Runoff

The voluminous river runoff from the Baltic Sea catchment areas is a significant contributor to the water budget of the Baltic Sea. In this section, the total river runoff is indirectly retrieved using Eq. (7) by considering the conservation of Baltic Sea water mass. Note the computed river runoff also includes groundwater flow into the Baltic basin, as we simplified this equation (cf. Eq. (1) in Omstedt et al., 2004).

Figure 10a shows the runoff computed from the original and corrected model. A fourth-order Butterworth low-pass filter

with cutoff frequency at 1/30 days⁻¹ was applied to remove undesirable high-frequency component of the runoff. The river runoff derived from the corrected model (blue line) varies in a range of from -1e5 to 1.2e5 m³/s. Positive runoff indicates the addition of flow to seawater, whereas negative runoff signifies the withdrawal of water from the sea. This withdrawal may result in temporary inflows of seawater into the river or estuarine system due to rising Baltic water level or storm surges. In this figure, the yellow area indicates the period during which the river runoff of the corrected model was computed to be

higher than that of the original model, while the light blue area indicates the opposite.



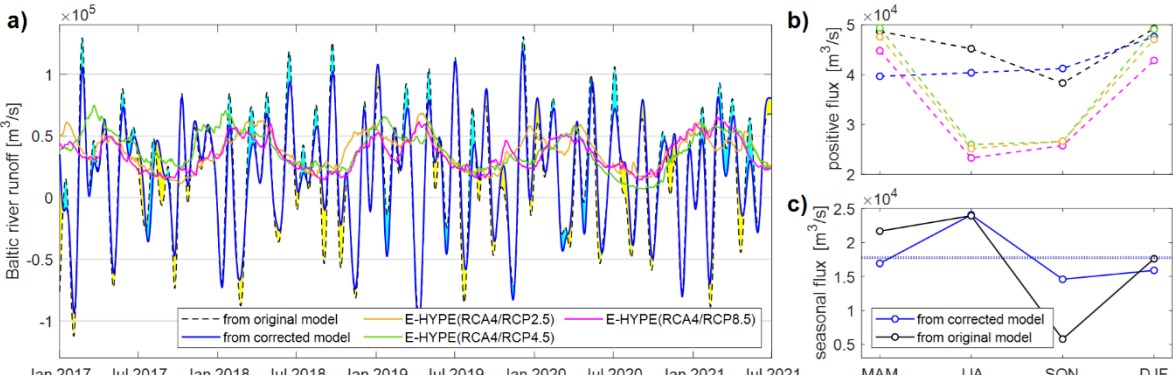

**Figure 10: River runoff computed from Eq. (7) using both original and corrected model: a) daily timescale, along with total river discharge obtained from E-HYPE dataset, b) seasonal averages of positive runoff (flux into the sea) and E-HYPE river discharge, c) seasonal average of the river runoff computed from original and corrected model.**

Comparison of different sources of the river discharge was also examined in Figure 10a, using the total river discharge computed from the E-HYPE dataset (Berg et al., 2021). In this figure, the total river discharge is determined by integrating the flux along the Baltic Sea boundary on a daily timescale. The Nemo-Nordic model is also forced by E-HYPE river discharge (Kärnä et al., 2021). Different version of the E-HYPE model was used with all of then showing a similar pattern. Comparison of the E-HYPE river discharge with our estimated river runoff shows more or less similar range in positive

values, with our results displaying a higher frequency of events and extreme runoff. Note, in forced hydrodynamic models, the water mass conservation inside the model domain is also regulated by the sea level at open boundary conditions. In addition, the Baltic river runoff estimated from the water balance calculation can differ from the river forcing used for the model for it is two different concepts of deriving river discharge. Based on our knowledge, the E-HYPE model does not specifically account for seawater withdrawal (i.e., reverse flow from sea to land). It is typically used to model the flow from

land to sea, as the model considers the sea basin as the downstream. In comparison with the computed runoff using Eq. (7), it can be observed that the presented approach is able to determine the flow in interactions between land and sea in more temporal detail due to accurate water balance computation. This estimation can potentially be coupled with a hydrological model.

Figure 10b shows the seasonal average of the positive fluxes, where a discrepancy can be observed between E-HYPE and the

presented approach in summer and autumn. To fully understand this, we refer to Section 3.4, with reference to Table 2 and figure 11, where it is explained that during summer, the Baltic basin experiences a low 'dynamic water volume' while its rate of change ($dV(t)/dt$) is positive and at its highest level. Also observe that the water volume increases in the following autumn season (see Figure 11a); this indicates that the most likely source of replenishment in the summer that the Baltic Sea experiences is due to river runoff. Therefore, it can be inferred (according to Figure 10b) that the E-HYPE dataset

underestimates river discharge in the summer months. In addition, since the original Nemo-Nordic utilized the E-HYPE model, this under estimation is also instilled in the original Nemo-Nordic model and is most visual in the autumn months,




where the hydrodynamic model compensates for its lost water volume by overestimating the Baltic inflow and underestimating the outflow (see net Baltic flow in Figure 8c).

The long-term monthly mean of Baltic river runoff ranges from 1e4 to 2.5e4 m$^3$/s (Leppäranta and Myrberg, 2009). In Figure
10c, it can be observed that the seasonal average of the computed runoff from the corrected model roughly follows the long-term mean. One can also observe that the original model has difficulties tuning water flow inputs in autumn, which leads to a substantial underestimation of the runoff.

### 3.4 Seasonal Water Budget

This section summarizes the computations of the Baltic Sea water balance on a seasonal timescale. The seasonal average of
dynamic water volume calculated using Eq. (2) is shown in Figure 11a. This figure shows that for spring, the volume was the lowest (78.9±60 km$^3$), whilst for autumn and winter the volume is the highest (120±58 km$^3$ and 124±80 km$^3$, respectively). The lowest standard deviation is for summer (42 km$^3$) when the basin experiences calm sea conditions. The corresponding values calculated from the original Nemo-Nordic model are indicated by dotted black lines.

Figure 11b and 11c demonstrate the seasonal variation of parameters described in Eq. (6) together in terms of seasonal
average and standard deviation for the original model (dashed lines) and corrected model (solid lines). The seasonal mean of the net atmospheric flux ($P + E$) is insignificant, with a range of 5×10$^{-3}$ km$^3$/day, and is therefore not included in this figure. The significant discrepancy in water volume rate occurred in winter (0.51 km$^3$/day) and spring (0.27 km$^3$/day). The seasonal variations of river runoff (shown in green lines) and Baltic inflow (shown in red lines) were discussed above and are also depicted in these figures. The standard deviation of the water volume rate indicates that the original model exhibits greater
variation than the corrected model (except in winter), likely because of the influence of river runoff, which follows a similar pattern. The standard deviation in the Baltic inflow/outflow (Figure 11c) remains consistent both before and after model correction.

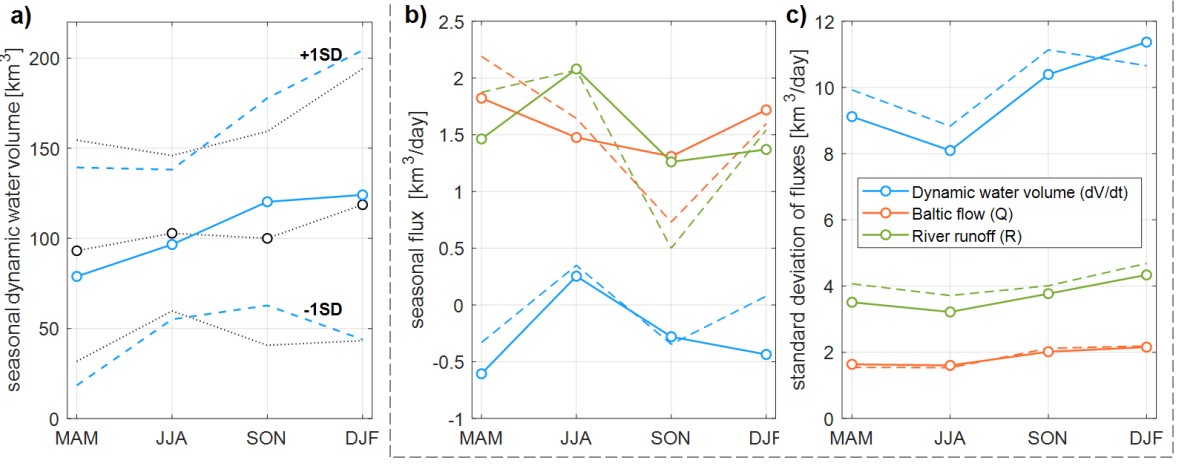





**Figure 11: Seasonal variability of dynamic water volume and the Baltic water mass fluxes. a) dynamic water volume computed**
**from the original model (dotted lines) and corrected model (solid lines). Average (b) and standard deviation (c) of the Baltic water**
**mass flux (Eq. 6), computed from daily time series. Solid lines and dashed lines represent the corrected model and the original**
**model, respectively.**

Figure 11 illustrates the discrepancy in the original model water volume ($V$) is compensated by river runoff ($R$) and Baltic

flow ($Q$). This figure indicates the balance of the Baltic input water mass flux, where the observations confirm the hourly

corrected model. Table 2 summarizes the computed values using the corrected DT from Figure 8, 10, and 11. On the

seasonal timescale, one can observe that the Baltic Sea has high river discharge rate in summer months compared to other

seasons. Thus, in summer, the primary source of replenishment in the Baltic Sea is river runoff, as the water exchange

through the Danish Straits is limited; the Baltic Sea still has a positive DT with respect to the Kattegat; and this season has

typically calm conditions and there is no dominant wind for water piling up to northern and eastern basins. This study shows

that a recursive analysis of the Baltic Sea equilibrium can help identify potential biases in the datasets and enhance model

performance.

**Table 2: Seasonal variation of the Baltic dynamic water volume and water budget components.**

|  | Spring | Summer | Autumn | Winter |
|---|---|---|---|---|
| **Dynamic Volume (km³)** | 78.88 (±60.4) | 96.61 (±41.5) | 120.30 (±57.5) | 124.12 (±80.2) |
| **Rate of Dynamic Volume (km³/day)** | -0.61 (±9.1) | 0.25 (±8.1) | -0.28 (±10.4) | -0.44 (±11.4) |
| **Barotropic Inflow (km³/day)** | -1.37 (±0.5) | -1.45 (±0.4) | -1.71 (±0.6) | -1.94 (±0.6) |
| **Barotropic Outflow (km³/day)** | 2.33 (±0.5) | 2.10 (±0.4) | 2. 28 (±0.5) | 2.71 (±0.5) |
| **Baltic Net flow (km3/day)** | 1.82 (±1.6) | 1.48 (±1.6) | 1.31 (±2.0) | 1.72 (±2.2) |
| **River Runoff (km³/day)** | 1.46 (±3.5) | 2.08 (±3.2) | 1.26 (±3.8) | 1.37 (±4.3) |

**4 Conclusions**

This study demonstrates that utilization of Dynamic Topography (DT), which is the deviation of sea level from a geoid

reference surface, offers new and accurate possibilities for quantifying and contributing to better understanding the most

relevant parameters such as spatial anomaly of DT, dynamic water volume and components of the Baltic Sea's water budget

(in particular Baltic inflow/outflow, variations in water volume, and river runoff rates).

Specifically, and most importantly, we first emphasize on calculating the dynamic water volume V(t), which represents the

water volume that varies over time on top of the constant water volume in the Baltic Sea. Quantification of the dynamic

water volume allows derivation of the Baltic flow rate through the Danish Straits (the main channel connecting the Baltic



Sea to the open ocean) and as a consequence we could compute and examine: (i) changes in the inflow/outflow of the Baltic Sea and also (ii) indirectly river discharge.

Results show on average a permanent positive dynamic water volume of 100 km³ relative to the global mean sea level reference marked at the NAP. The volume typically decreases by $78.9 \pm 60$ km³ during the spring, and increases by $121 \pm 57$ km³ and $124 \pm 80$ km³ during the autumn and winter months, respectively. This seasonal trend exists for all the sub-basins. Furthermore, by deriving spatial anomaly of DT, we can examine the variation of each sub-basin relative to each other. It is known that a permanent density gradient exists between the northern and southern sub-basins, and this causes a sea level difference between these two basins. The calculated DT anomalies also showed a strong anti-correlation between north and south regions with it being a maximum during autumn and winter months (maximum difference of 32 cm). Also, atmospheric drivers (especially winds) were found to influence anomaly of DT in the sub-basins. In the Baltic Proper, the anomaly of DT approach around 0 cm, this indicates that this region closely represents an equilibrium mean DT for the entire Baltic Sea (Figure 7). Indicating in terms of a monitoring aspect that any significant changes in DT values in this area may be used as climatic change indicator or of changing ocean dynamics of the Baltic Sea (e.g. Baltic inflows).

Examination of the spatial anomaly of DT and its standard deviation can also potentially identify the areas that may be most affecting by increasing sea levels, extremes and coastal erosion. In our study sensitive areas such as Gulf of Finland, Gulf of Riga and southern Baltic Sea were identified which was also similar to that identified in other studies (Weisse et al., 2021; Pindsoo and Soomere, 2020). Therefore, it can be inferred that sea level rise in this basin, coupled with changes in the wind pattern and ice formation, could potentially increase the frequency and intensity of extreme sea levels, especially in the northern and eastern regions of the Baltic Sea (Weisse et al., 2021).

The water exchange between the Baltic Sea and the North Sea represents a vital process that discharges and replenishment waters of the Baltic Sea. By comparing the DT of the entire Baltic Sea with that of the Kattegat, the barotropic flow through the Danish Strait was calculated. Based on the criterion of events lasting at least one day classification of inflow and outflow events were performed. Our results confirm that due to the positive freshwater budget of the Baltic Sea there is basically a constant outflow of water through the Danish Strait with rates average of 2.36 km³/day for outflow and 1.6 km³/day for inflow (when they occur). Thus, during the time frame of this study, the maximum volume of imported saline water from North Sea was 22.5 km³. In addition, five inflow events per year with a duration of more than 5 days was observed. Major Baltic Inflow was not observed during our study; thus, the examined events were mostly that of medium and small scale. A seasonal pattern was observed with highest inflows mainly occurring in the autumn and winter months and highest outflows occur in the winter months. As a result, the net Baltic flow, with the maximum positive rate occurring in spring and winter, and the minimum positive rate in autumn.

It was demonstrated that for some of the inflow events, salty bottom water reached as far as the southern Baltic Sea but not as far as the Baltic Proper. This has previously been known to occur for small and medium flows (Sellschopp et al., 2006). Additionally, most intriguing was that prior to intense inflow events, an increase in bottom salinity anomaly was observed in the Southern Baltic Sea basin yet the Bornholm basin shows a decrease (Figure 9). This may be due to the influence of winds



and mixing, however, to fully understand this further examination and data sources are required, which can be examined for future studies.

River runoff flux was indirectly derived, with results showing maximum values of approximately 2.1 km$^3$/day in summer. This rate decreases to below 1.5 km$^3$/day for other seasons. These rates are similar to that found in other studies (Leppäranta and Myrberg, 2009). A comparison with the E-HYPE dataset showed that this dataset seems to underestimate river discharge flux during summer and autumn. This leads to the original HDM compensating for the input flows by over/underestimating Baltic in/outflows during the autumn months. This research suggests that the presented approach of deriving river runoff could enhance our understanding of hydrological models and improve the accuracy of river discharge modelling.

In this study, some simplifications have been made, especially in Eq. (6), which the complete form of the equation should contain volume change due to ice advection, thermal expansion and salt contraction, groundwater inflow, and the volume change due to vertical land motion in Baltic Sea (Omstedt et al., 2004). However, we assume that the corrected DT, and subsequently the dynamic water volume based on its geoid-referenced definition, explains these changes in water volume, and the groundwater inflow was merged with river runoff for simplification. This can be an advantage for the indirect approach used in calculating river discharge flow. In addition, an empirical equation (Eq. 5) was used for estimating Baltic inflow/outflow; and we have mainly considered barotropic flows, when baroclinic flows can also occur in the Baltic sea especially during the summer months (Feistel et al., 2006; Mohrholz, 2018).

The method and results of this study demonstrates that utilizing of a geoid referred DT can significantly contribute to quantification and better understanding of the marine dynamics of the Baltic Sea. The sea level dynamics of this complex and sensitive sea area can greatly contribute to understanding the processes governing the hydrodynamics. Such insights are important for informing sustainable management practices for marine resources and for developing effective mitigation and adaptation strategies that assists the adverse effects of climate change on the Baltic Sea ecosystem. In addition, accurate sea level measurements can improve forecasting skills, particularly in predicting extreme events, which have gained significant attention due to the effects of climate change.

Note climate change impacts usually require a longer time series of data, but this study focuses on quantifying the seasonal characteristics of the water budget components. The method employed can also be used for longer time series and may have beneficial use for climate studies (Hordoir et al. 2015; Meier et al., 2023).

**Competing interests**

The contact author has declared that none of the authors has any competing interests.



**Acknowledgments**

The research is supported by the Estonian Research Council grants PRG1785 "Development of continuous dynamic vertical reference for maritime and offshore engineering by applying machine learning strategies" and PRG1129.

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
