# Peer review of "Quantification of Baltic Sea Water Budget components Using Dynamic Topography"

_EGUsphere, 2024_

## Referee Comment (RC1)

The manuscript presents an interesting study of the water balance of the Baltic Sea. The authors are continuing their previous work defining the dynamic topography of the Baltic Sea and using the gathered information to draw some conclusions on the dynamics of the basin. The study seems to be done rigorously and there are no evident problems with it. The biggest drawback is the logic of the text, as it is organized in somewhat confusing manner, some more explanation below.

The introduction is somewhat hard to follow. The general explanation of Baltic Sea characteristics, previous studies and authors' own previous results are not explained in a very logical order. The "traditional" logic to an introduction is "general description – what has been done before – what has been missed – what are our solutions to the problem".

Also in the following chapters the text jumps between what will be done in the current study and what has been done before, sometimes not really explaining the previous results (e.g. RefBias). There are hints to coming results, general explanations and even discussion on the results in the methods part. Chapter 3 should be renamed to results and discussion, as there is not separate discussion chapter in the manuscript. Some of the figures are a bit too busy, e.g. Figure 2 (see below) as well as time series plots showing different basins and happenings (e.g. Figures 5, 8, 9).

The methods used in the manuscript as well as the results and conclusions drawn from them seem to be rigorous and show no problems. With reorganizing the text and clarifying some details the manuscript should be publishable, although with my expertise I am not comfortable to assess the importance of the results from oceanographical aspect.

Some detailed comments:

Line 30-32, p. 1-2 "(iii) limited vertical mixing, as convection, mechanical mixing, entrainment, and advection are known to occur for tidal amplitudes are more or less small." what does this latter part mean? What "are known"?

Line 62, p. 3 "They" at the beginning of the line. Does this refer to HDMs? Or sea level component, or what?

Line 120, p. 5 "a constant geopotential Was its implicit vertical" ? Typo?

Line 122, reference to RefBias is a bit confusing. The RefBias is not used later in the text, why do you want to introduce this term at all?

Line 153-154, p. 6, Eq. (1) the u and v are not explained. They are the geostrophic current's two components, right?

Line 160, p. 6. The chapter starting with "in this study" is a bit out-of-place, and the whole thing is a bit of a mess, own results, others' work, what?

Figure 2 is very hard to decipher with inset without borders, with "seasonal" = winter, spring, summer, autumn?, too many things, blue and red colours?

Line 215, p. 9, should probably be in the discussion?

Lines 257-260, p. 11, The first two sentences seem to explain the same thing, but not the same thing. It is a bit confusing, please check.

Line 427, p. 18, CTD?

Line 542, p. 23 "…were identified … that identified…" Not very good language.

Line 546, p. 23 "replenishment" should this be a verb: replenishes?

---

## Author Comment (AC1)

**Response to Reviewer #1**

The reviewer's comments are highlighted in *blue italic fonts*, while the authors' responses are presented in black regular fonts. Modified texts of the manuscript are presented in *black italic fonts*.

*The manuscript presents an interesting study of the water balance of the Baltic Sea. The authors are continuing their previous work defining the dynamic topography of the Baltic Sea and using the gathered information to draw some conclusions on the dynamics of the basin. The study seems to be done rigorously and there are no evident problems with it. The biggest drawback is the logic of the text, as it is organized in somewhat confusing manner, some more explanation below.*

We appreciate your valuable comments and the time you dedicated to reviewing our manuscript. In the following, we address each of your comments and improve the manuscript in the revised version.

*The introduction is somewhat hard to follow. The general explanation of Baltic Sea characteristics, previous studies and authors' own previous results are not explained in a very logical order. The "traditional" logic to an introduction is "general description – what has been done before – what has been missed – what are our solutions to the problem".*

Thanks for this remark. In the introduction, we discuss "what we have done in our previous work" (that is a part of our solution), "what has been missed and our solution", "general description of the current study and what has been done before", and "the paper structure". We reorganize and modify the introduction text of L40 to L79 to make it more logical, as follows:

*"In our previous study, we demonstrated that utilizing a geoid surface (i.e., an equipotential surface of the earth) as a reference surface for various sea level sources allows us to determine accurate sea level variation (Jahanmard et al., 2022 and 2023a). The sea level variation with respect to the geoid surface is referred to as ocean dynamic topography (DT). Referencing sea level to a stable reference surface provides opportunities for further studies, such as developing data-driven sea level forecasting (Rajabi-Kiasari et al., 2023) or quantifying the components of the Baltic Sea water budget, which will be discussed in this study.*

*Previously, determination of accurate DT for both coastal and offshore areas has been limited due to access of an accurate high-resolution geoid model. The Baltic Sea countries, however, have developed the NKG 2016 geoid model, whereas presently the geoid-based Baltic Sea Chart Datum BSCD2000 vertical datum (Liebsch et al., 2023) is being implemented in the Baltic Sea countries. Adopting the geoid model is suitable to transfer all the sea level datasets to the same zero level that is long-term mean sea level marked at Normaal Amsterdams Peil (NAP), allowing thus seamless sea level data (e.g. Hydrodynamic Model, tide gauge, and satellite altimetry) integration from coast and offshore.*

*Ideally, the sea level component in hydrodynamic models (HDM) can be employed for assessing the water budget. However, HDMs are limited by modelling errors and a vertical reference bias that constitutes altogether as an overall bias which varies both spatially and temporally. This often prevents the link with other sea level data sources (in-situ and satellite data). In the Baltic Sea, different methods such as interpolation and deep learning were employed to examine the bias (Jahanmard et al., 2021; 2022; 2023a). In this study, we use the corrected HDM from Jahanmard et al. (2023a) along with the original model. In the corrected HDM, low-frequency errors relative to observations were reduced, resulting in an improvement in the model's RMSE from an average of 7*

*cm to 4 cm. Additionally, the vertical reference bias between the model and observations was removed by 18.1 cm. In the following, we demonstrate that the corrections made to the original HDM preserve its general dynamics while improving overall performance. This is summarized in Section 2.1, followed by the validation of the geostrophic currents in the corrected models. The use of DT enables the calculation of dynamic water volume relative to a physically meaningful reference surface, which is later used in the calculations of the water budget variations and barotropic flows in the following sections.*

*Regional variations in sea level and coastline dynamics within the Baltic Sea provide a valuable framework for investigating climate variability, extreme events, and processes with global significance, supported by extensive historical and instrumental records (Harff et al., 2017; Weisse et al., 2021). Sea level variability in the Baltic Sea can be categorized into processes that alter the total Baltic water volume and that redistribute water within the Baltic Sea (Samuelsson and Stigebrandt, 1996). Timescales of about half a month or longer predominantly influence changes in the total water volume of the Baltic Sea. Conversely, shorter-term processes, constrained by the limited transport capacity across the Danish Straits, primarily result in the redistribution of water within the basin (Johansson, 2014; Soomere et al., 2015; Weisse et al., 2021). The long-term variability of the Baltic Sea water budget is influenced by basin-averaged mean sea level rise—primarily driven by the influx of mass from the adjacent North Sea as an external signal, with minor contributions from basin-internal water mass redistribution due to local baroclinicity (Gräwe et al., 2019)—and by crustal deformation in the Baltic region caused by postglacial uplift (Richter et al., 2012). The multitude of processes contributing to sea level variations in the Baltic Sea complicates the interpretation of its dynamics (Weisse et al., 2021). Hence, this study aims to quantify the Baltic Sea water budget by adopting a common stable reference surface for sea level modeling and observations, which enables accurate determination of sea level dynamics.*

*Therefore, utilization of DT allows us to: (i) examine the dynamic water volume distribution and its seasonal and sub-basin variations; (ii) quantify barotropic inflow and outflow through the Danish Strait; and (iii) estimate river runoff by using the Baltic Sea water balance computation (Omstedt et al. 2004; Reckermann et al., 2011)."*

{Added references}:

*Harff, J., Deng, J., Dudzińska-Nowak, J., Fröhle, P., Groh, A., Hünicke, B., ... & Zhang, W. (2017). What determines the change of coastlines in the Baltic Sea?. Coastline changes of the Baltic Sea from south to east: Past and future projection, 15-35.*

*Samuelsson, M., & Stigebrandt, A. (1996). Main characteristics of the long-term sea level variability in the Baltic sea. Tellus a, 48(5), 672-683.*

*Johansson, M. (2014). Sea level changes on the Finnish coast and their relationship to atmospheric factors.*

*Soomere, T., Eelsalu, M., Kurkin, A., & Rybin, A. (2015). Separation of the Baltic Sea water level into daily and multi-weekly components. Continental Shelf Research, 103, 23-32.*

*Gräwe, U., Klingbeil, K., Kelln, J., & Dangendorf, S. (2019). Decomposing mean sea level rise in a semi-enclosed basin, the Baltic Sea. Journal of Climate, 32(11), 3089-3108.*

*Richter, A., Groh, A., & Dietrich, R. (2012). Geodetic observation of sea-level change and crustal deformation in the Baltic Sea region. Physics and Chemistry of the Earth, Parts A/B/C, 53, 43-53.*

*Also in the following chapters the text jumps between what will be done in the current study and what has been done before, sometimes not really explaining the previous results (e.g. RefBias). There are hints to coming results, general explanations and even discussion on the results in the methods part. Chapter 3 should be renamed to results and discussion, as there is not separate discussion chapter in the manuscript. Some of the figures are a bit too busy, e.g. Figure 2 (see below) as well as time series plots showing different basins and happenings (e.g. Figures 5, 8, 9).*

Thanks for your comments and we understand that it may appear a bit confusing. Our logic for Section 2 (Methods) was that we thought it would first be useful to summarize important aspects of the research we have done before and presented an additional validation in Section 2.1, as this was important to understand the overall methodology employed in the present study. The term of "RefBias" is now corrected and replaced by reference bias (please refer to the following comments). This is then followed by the remaining sections (section 2.2, 2.3, and 2.4) that all describe the methodology employed in the present study. So, in summary what is explained in the method section (more explicitly in Section 2.1) is a brief continuation of our previous work and its link to the current study, which is not a part of the current study result. Since the current paper is presented as an application of the geoid-referenced sea level determination, section 2.1 is essential as background knowledge to understand the methodology explained in sections 2.2, 2.3, and 2.4.

Thank you for your suggestion. Chapter 3 is renamed to "Results and Discussion". We will improve Figure 2 (please refer to the following comments). For time series plots, we group different basins and happenings together to visually observe their relative variations and events; and also avoid inserting several figures, which could make the manuscript lengthier. We could not identify any specific aspects for improvement in Figure 8 and 9 (nevertheless, any further suggestions in this regard are very welcome). Figure 5a is modified as follows, by removing the boxes and highlighting them on time axis.

[Figure]

*The methods used in the manuscript as well as the results and conclusions drawn from them seem to be rigorous and show no problems. With reorganizing the text and clarifying some details the manuscript should be publishable, although with my expertise I am not comfortable to assess the importance of the results from oceanographical aspect.*

Thanks for your constructive comments and feedback. We address them as explained above and in the following detailed comments.

***Some detailed comments:***

*Line 30-32, p. 1-2 "(iii) limited vertical mixing, as convection, mechanical mixing, entrainment, and advection are known to occur for tidal amplitudes are more or less small." what does this latter part mean? What "are known"?*

Thanks for your remark. We correct the text as follows for clarity:

*"(iii) limited vertical mixing, as processes such as convection, mechanical mixing, entrainment, and advection are known to be limited when tidal amplitudes are relatively small."*

*Line 62, p. 3 "They" at the beginning of the line. Does this refer to HDMs? Or sea level component, or what?*

It refers to HDMs. We replace the word "*They*" with "*HDMs*".

*Line 120, p. 5 "a constant geopotential Was its implicit vertical" ? Typo?*

Thanks for your remark. Yes, there should be a space after *"W":*

*"a constant geopotential W as its implicit vertical"*

*Line 122, reference to RefBias is a bit confusing. The RefBias is not used later in the text, why do you want to introduce this term at all?*

Thanks for this remark. This term had been used in initial versions of the paper, but it was removed in iterations. We modify L122 as follows:

*"which was quantified by reference bias"*

*Line 153-154, p. 6, Eq. (1) the u and v are not explained. They are the geostrophic current's two components, right?*

True. These two terms represent the geostrophic components in x and y directions. Text is updated as follows:

*"Surface geostrophic current ($u_g$ and $v_g$ in the x and y directions, respectively), which represents the balance between pressure gradients and the Coriolis force, can be deduced from the following determination:"*

*Line 160, p. 6. The chapter starting with "in this study" is a bit out-of-place, and the whole thing is a bit of a mess, own results, others' work, what?*

Thanks for your remark. This chapter (L160 to L182) discusses Figure 2 and compares the calculated surface geostrophic circulation from the corrected HDM with other studies. The text is modified and condensed as follows for clarity:

*"Figure 2 shows seasonal surface geostrophic currents computed from the corrected HDM. For this calculation, we use 7-point stencils (Arbic et al., 2012). A similar computation with the original HDM shows*

*the same geostrophic pattern. This pattern agrees well with the Baltic circulation computed from other studies (Döös et al., 2004; Soomere and Quak, 2013; Placke et al., 2018; Hinrichsen et al., 2018; Barzandeh et al., 2024). As observed in Figure 2, a cyclonic circulation exists in the Baltic Proper (Jędrasik and Kowalewski, 2019; Liblik et al., 2022), intensifying during the autumn and winter months. Furthermore, there is a cyclonic circulation in the Bothnian Sea that also strengthens in the autumn and winter months. In the Gulf of Finland, a persistent westward current is along the Finish coast and a narrow eastward current along the Estonian coast (Alenius et al., 1998; Soomere et al., 2011).*

*Therefore, this figure shows that the HDM correction not only improves the accuracy of the simulated sea surface, in comparison with satellite altimetry and tide gauge data (presented in Jahanmard et al., 2023a), but also preserved the underlying surface geostrophic circulation patterns over the Baltic Sea."*

*Figure 2 is very hard to decipher with inset without borders, with "seasonal" = winter, spring, summer, autumn?, too many things, blue and red colours?*

Thanks for your comment. We agree that this figure is too decorated with various things. Therefore, we remove the figure and replace it with only seasonal geostrophic currents computed from the corrected HDM. The same computations with the original HDM show the same seasonal pattern, which is what we wanted to express here. To avoid repeating figures, we only present the figure of the corrected HDM. Accordingly, the text is also modified as presented in the previous comment.

[Figure]

*Figure 2: Seasonal surface geostrophic currents for the period of 2017.0 to 2021.5.*

*Line 215, p. 9, should probably be in the discussion?*

This sentence is related to Figure 3 and provides a sense of the values represented in this figure. Since this figure was used to express the problem of this study, we prefer to keep it in the Method section.

Lines 257-260, p. 11, The first two sentences seem to explain the same thing, but not the same thing. It is a bit confusing, please check.

Thanks for pointing this out. The sentences are modified as follows:

*"The saline water exchange between the Baltic Sea and the North Sea typically occurs somewhere between the Kattegat and the Danish Straits, which varies depending on the prevailing baroclinic and barotropic forcing conditions."*

Line 427, p. 18, CTD?

Thanks, it is corrected by adding *"CTD (conductivity, temperature, and depth)"* to that line.

Line 542, p. 23 "…were identified … that identified…" Not very good language.

Thanks. This sentence is modified as follows:

*"In our study, we identified sensitive areas, including the Gulf of Finland, the Gulf of Riga, and the southern Baltic Sea. These findings align with those reported in other studies (..''*

Line 546, p. 23 "replenishment" should this be a verb: replenishes?

Thanks for your observations. It is corrected to *"replenishes"*

---

## Author Response (AR1)

Revised R1 version, submitted, 18.02.2025
**"Quantification of Baltic Sea Water Budget components Using Dynamic Topography"**
Responses to reviewers' remarks

**General**

Dear Sir/Madam,
First, we would like to thank both reviewers for their constructive comments, valuable suggestions, and the time they dedicated to reviewing our manuscript. We will respond to the comments and address the reviewer's concerns in our reply. In the following, the reviewer's comments are presented in *blue italics*, while our responses appear in black regular font. When possible, modified text is highlighted in *black italics* along with the corresponding line number (according to the original manuscript). Otherwise, please refer to the revised version or the edited manuscript in track changes mode. We have slightly adjusted our response to the first reviewer (with respect to our online reply) based on the final revision. We hope you find our revised version suitable for publication in Ocean Science and look forward to hearing from you in the near future.

Sincerely,
Vahidreza Jahanmard
On behalf of the co-authors

**REVIEWER #1**

*The manuscript presents an interesting study of the water balance of the Baltic Sea. The authors are continuing their previous work defining the dynamic topography of the Baltic Sea and using the gathered information to draw some conclusions on the dynamics of the basin. The study seems to be done rigorously and there are no evident problems with it. The biggest drawback is the logic of the text, as it is organized in somewhat confusing manner, some more explanation below.*

We appreciate your valuable comments and the time you dedicated to reviewing our manuscript. In the following, we address each of your comments and improve the manuscript in the revised version.

*The introduction is somewhat hard to follow. The general explanation of Baltic Sea characteristics, previous studies and authors' own previous results are not explained in a very logical order. The "traditional" logic to an introduction is "general description – what has been done before – what has been missed – what are our solutions to the problem".*

Thanks for this remark. In the revised version, we have modified the introduction and the structure of the manuscript. Additionally, the relevant background information from our previous study has been moved to Section "2. Background and Dataset". We reorganize and modify the introduction text of L40 to L76 to make it more logical, as follows:

*Ideally, the sea level component in hydrodynamic models (HDM) can be employed for assessing the water budget. However, HDMs are limited by modelling errors and a vertical reference bias that constitutes altogether as an overall bias which varies both spatially and temporally. This often prevents the link with other sea level data sources (in-situ and satellite data). In the Baltic Sea, methods such as interpolation and deep learning were used to examine the bias by employing a common geoid-based vertical reference surface for data sources, through which absolute ocean dynamic topography (DT) was determined (Jahanmard et al., 2022; 2023a). The use of DT enables the calculation of dynamic water volume relative to a physically meaningful reference surface, which is later used in the calculations of the water budget variations and barotropic flows in the following sections.*

*Regional variations in sea level and coastline dynamics within the Baltic Sea provide a valuable framework for investigating climate variability, extreme events, and processes with global significance, supported by extensive historical and instrumental records (Harff et al., 2017; Weisse et al., 2021). Sea level variability in the Baltic Sea can be categorized into processes that alter the total Baltic water volume and that redistribute water within the Baltic Sea (Samuelsson and Stigebrandt, 1996). Timescales of about half a month or longer predominantly influence changes in the total water volume of the Baltic Sea. Conversely, shorter-term processes, constrained by the limited transport capacity across the Danish Straits, primarily result in the redistribution of water within the basin (Johansson, 2014; Soomere et al., 2015; Weisse et al., 2021). The long-term variability of the Baltic Sea water budget is influenced by basin-averaged mean sea level rise—primarily driven by the influx of mass from the adjacent North Sea as an external signal, with minor contributions from basin-internal water mass*

*redistribution due to local baroclinicity (Gräwe et al., 2019)—and by crustal deformation in the Baltic region caused by postglacial uplift (Richter et al., 2012). The multitude of processes contributing to sea level variations in the Baltic Sea complicates the interpretation of its dynamics and inflows (Weisse et al., 2021). Therefore, quantifying the dynamics of the water column within the Baltic Sea and its interactions with the North Sea and river inflows provides important insights into regional hydrodynamics and water exchange processes.*

*Also in the following chapters the text jumps between what will be done in the current study and what has been done before, sometimes not really explaining the previous results (e.g. RefBias). There are hints to coming results, general explanations and even discussion on the results in the methods part. Chapter 3 should be renamed to results and discussion, as there is not separate discussion chapter in the manuscript. Some of the figures are a bit too busy, e.g. Figure 2 (see below) as well as time series plots showing different basins and happenings (e.g. Figures 5, 8, 9).*

Thanks for your comments and we understand that it may appear a bit confusing. Our logic for Section 2 (Methods) was that we thought it would first be useful to summarize important aspects of the research we have done before and presented an additional validation in Section 2.1, as this was important to understand the overall methodology employed in the present study. However, we agree that this section requires further consideration. We have modified and restructured the text according to your suggestions as follows:

- Section 2.1 is deleted and moved to section "2. Background and Dataset" to explain the background of this study. Accordingly, one will be added to the number of the following sections (e.g., "3. Method"). Also, some background information from the introduction has been merged into this section.
- Datasets are introduced in Section "Background and Dataset". For instance: L291-293; L425-429; L460-466. Therefore, we can delete Figure 9a to make Figure 9 less cluttered.
- Figures 3 and 4 are moved to the Section 3.1 and renamed it to "Dynamic Water Volume and Co-oscillation of Sub-Basins". Subsequently, we have minor modifications in this section. Therefore, we have kept all results in Section 4.
- "3. Results" section is renamed to "4. Results and Discussions", and information regarding datasets is transferred to "Background and Dataset".
- We have also made minor textual edits to other parts of the manuscript for better readability (please refer to tracking mode of the revised version)
- The term of "RefBias" is deleted.
- Figure 2 is replaced with the following figure (shown in the relevant comment below) to demonstrate that our correction (results of our previous work) does not affect the underlying Baltic Sea circulation pattern.
- For the time series plots, we have grouped different basins and events together to visually observe their relative variations, while also avoiding the inclusion of multiple figures, which would lengthen the manuscript. As a result, Figure 5a has been modified by removing the boxes and highlighting them on the time axis, making it less cluttered. Also, Figure 9a was removed, as mentioned above. Figure 8 remains unchanged, as we

did not find any suitable corrections; however, any further suggestions in this regard are welcome.

*The methods used in the manuscript as well as the results and conclusions drawn from them seem to be rigorous and show no problems. With reorganizing the text and clarifying some details the manuscript should be publishable, although with my expertise I am not comfortable to assess the importance of the results from oceanographical aspect.*

Thanks for your constructive comments and feedback. We address them as explained above and in the following detailed comments.

*Some detailed comments:*

*Line 30-32, p. 1-2 "(iii) limited vertical mixing, as convection, mechanical mixing, entrainment, and advection are known to occur for tidal amplitudes are more or less small." what does this latter part mean? What "are known"?*

Thanks for your remark. We correct the text as follows for clarity:

L31: *(iii) limited vertical mixing, as processes such as convection, mechanical mixing, entrainment, and advection are known to be limited when tidal amplitudes are relatively small.*

*Line 62, p. 3 "They" at the beginning of the line. Does this refer to HDMs? Or sea level component, or what?*

Thanks. It refers to HDMs. We replace the word "*They*" with "*HDMs*".

*Line 120, p. 5 "a constant geopotential Was its implicit vertical" ? Typo?*

Thanks for your remark. Yes, there should be a space after *"W"* that represents the constant geopotential*:*

L120: *"a constant geopotential W as its implicit vertical"*

*Line 122, reference to RefBias is a bit confusing. The RefBias is not used later in the text, why do you want to introduce this term at all?*

Thanks for this remark. This term had been used in initial versions of the paper, but it was removed in iterations. It is removed from the text.

*Line 153-154, p. 6, Eq. (1) the u and v are not explained. They are the geostrophic current's two components, right?*

True. These two terms represent the geostrophic components in x and y directions. Eq. (1) has been removed through the revision and restructuring of the manuscript.

*Line 160, p. 6. The chapter starting with "in this study" is a bit out-of-place, and the whole thing is a bit of a mess, own results, others' work, what?*

Thanks for your remark. Given the revision made, this chapter (L160 to L182) is modified and summarized as follows:

(location in the revised manuscript: Section 2, before Figure 2): *Figure 2 shows the surface geostrophic currents computed from the corrected HDM. This figure confirms that the corrected HDM aligns with the quasi-steady circulation patterns observed in the Baltic Sea (Döös et al., 2004; Soomere and Quak, 2013; Placke et al., 2018; Hinrichsen et al., 2018; Barzandeh et al., 2024). Surface currents in the Baltic Sea are influenced by sea surface tilt, wind stress at the sea surface, and thermohaline horizontal gradient of density steered by Coriolis acceleration, topography, and friction (Leppäranta and Myrberg, 2009; Soomere et al., 2011). As a result, this figure indicates that our DT correction approach not only improves the accuracy of sea surface determination by integrating model data and observations but also preserves the underlying circulation patterns.*

*Figure 2 is very hard to decipher with inset without borders, with "seasonal" = winter, spring, summer, autumn?, too many things, blue and red colours?*

Thanks for your comment. We agree that this figure is too decorated with various things. Therefore, we remove the figure and replace it with only seasonal geostrophic currents computed from the corrected HDM. The same computations with the original HDM show the same seasonal pattern, which is what we wanted to express here. To avoid repeating figures, we only present the figure of the corrected HDM. Accordingly, the text is also modified as presented in the previous comment.

[Figure]

*Figure 2: Seasonal surface geostrophic currents computed form the corrected HDM for the period of 2017.0 to 2021.5.*

Thanks. This sentence is related to Figure 3 and provides a sense of the values represented in this figure. This figure and the explanations are moved to the "Results and Discussion" section.

*Lines 257-260, p. 11, The first two sentences seem to explain the same thing, but not the same thing. It is a bit confusing, please check.*

Thanks for pointing this out. The sentences are modified as follows:

L257: *The saline water exchange between the Baltic Sea and the North Sea typically occurs somewhere between the Kattegat and the Danish Straits, which varies depending on the prevailing baroclinic and barotropic forcing conditions.*

Thanks, corrected by adding *"CTD (conductivity, temperature, and depth)"* to that line.

*Line 542, p. 23 "...were identified ... that identified..." Not very good language.*

Thanks. Modified as follows:

L542: *In this study, we identified sensitive areas, including the Gulf of Finland, Gulf of Riga and southern Baltic Sea. These findings align with those reported in other studies (Weisse et al., 2021; Pindsoo and Soomere, 2020).*

*Line 546, p. 23 "replenishment" should this be a verb: replenishes?*

Thanks for your observations. Corrected.

*Like the other reviewer, I found this manuscript difficult to follow, mainly because of how it was structured. For example, there are many instances in Sections 2 and 3 where background information on the Baltic Sea sea level and known mass/volume transports are presented for the first time. This background should all go in Section 1 as motivation for the exact problem the authors are investigating. Then Section 2 should be on specific methods used, and Section 3 on an analysis of the results and comparing them to previous studies. Section 3 is not the time to introduce background/motivation information, as it distracts from the actual results.*

First, we would like to thank you for your comments, suggestions, and the time you dedicated to reviewing our manuscript. In the following, we will answer the comments and improve the revised version of the manuscript accordingly. We appreciate your concern about the structure of the manuscript, and we improve the text in the revised version as follows:

- Section 2.1 is deleted and moved to section "2. Background and Dataset" to explain the background of this study. Accordingly, one will be added to the number of the following sections (e.g., "3. Method"). Also, some background information from the introduction has been merged into this section.
- Datasets are introduced in Section "Background and Dataset". For instance: L291-293; L425-429; L460-466; deleting Figure 9a.
- "Results" section is renamed to "Results and Discussions", and information regarding datasets is transferred to "Background and Dataset".
- Figures 3 and 4 are moved to the Section 3.1 and renamed it to "Dynamic Water Volume and Co-oscillation of Sub-Basins". Subsequently, we have minor modifications in this section.
- We have also made minor textual edits to other parts of the manuscript for better readability (please refer to tracking mode of the revised version)

*The authors make a big deal (multiple time throughout the paper) that'll theirs is the first to do this with a model and observations related to a geoid. Okay. But please, this only needs to be stated once in the introduction and you don't have to keep repeating it. It is distracting!*

Thanks for your observations. The repeated sentences are removed, and we keep the relevant explanation in "Background and Dataset" section. For instance, these lines are deleted/modified to avoid repeating (*for convenience, we also include the corresponding line numbers in the clean version inside parentheses*): L80 (*L65*); L201-202 (*L171*); L207 (*L173*); L226-227 (*L176*); L278-279 (*L203*); L291-293 (*moved to Sec. 2; L138*); L314-315 (*L240*).

*I really do not see the point of section 2.1, comparing geostrophic currents. First of all, this is mixing methods and results. Second, as far as I can tell, this has nothing to do with the motivation of the paper (looking for overall mass transports into/out of the Baltic). It seems this is mainly to further motivate that their geoid-referenced model DT is good, but this has been demonstrated in other papers. While there is nothing wrong with the analysis, it is distracting from the main goals of this manuscript. I would cut this completely.*

Thanks for your comment. We agree that this section needs to be modified. We removed this section from the "Method" section because it explains the background and has mistakenly leaked into this section. As explained above, we added one more section to explain the previous study, background and datasets. Please refer to the revised version.

*I also do not see the point of Section 3.1 -- yes, the DT varies differently in different sub-basins. But how is this relevant to the overall goals of the paper, to assess transport between the Baltic and North Sea, and to estimate river inflow from the DT variations? It feels like it is tacked onto the study, but is not really motivated in Section 1.*

Thanks again for this comment. The internal dynamics of the Baltic Sea are also our interest in this manuscript. The DT anomaly in sub-basins reflects the internal dynamics of the Baltic Sea and the variation/co-oscillation of water volume within the sub-basins. While this section may not directly be related to the assessment of the overall water transport between the Baltic and North Sea or river inflows, it provides valuable insight into the co-oscillations among the sub-basins and their seasonality. This understanding is particularly relevant during inflow and outflow events, as it helps characterize the Baltic Sea's response to external forcing. We believe this section adds value to the overall analysis. Thus, we would like to modify this section as follows:

- Title is change to "Dynamic Water Volume and Co-oscillation of Sub-Basins"
- Figures 3 and 4 (results that were wrongly presented in section 3.2) are moved to this section.
- Modifying the text of the introduction to clarify our motivation, for example:

L76: … *The multitude of processes contributing to sea level variations in the Baltic Sea complicates the interpretation of its dynamics and inflows (Weisse et al., 2021). Therefore, quantifying the dynamics of the water column within the Baltic Sea and its interactions with the North Sea and river inflows provides important insights into regional hydrodynamics and water exchange processes.*

L104-106: *This study aims to use DT of the corrected HDM for 2017-2021.5 to quantify: (i) the dynamic water volume of the Baltic Sea; (ii) the seasonal and spatial distribution of DT anomalies, which represents internal dynamics and the variation/co-oscillation of water volume within the Baltic sub-basins; (iii) barotropic water exchange between the Baltic and North Seas; and (iv) total river runoff to the Baltic Sea using the water budget equation.*

*Finally, I have one methodology concern. It is not at all clear to me (although it may be buried in the introductory material in Section 3) that the authors have considered annual and low frequency steric (density) variations that contribute to the volume changes. The volume balance equation being used is based on a mass balance (barotropic) approach, where the density is unchanging (e.g., basically freshwater density). The authors emphasize the north-south mean density gradients in the Baltic, but do these change in time? Have the authors quantified if this is small enough to ignore, or is it even included in their model. My concern is that some of the volume changes are not actually related to the water fluxes into and out of the Baltic, but changes in the density structure.*

We appreciate your concern regarding the steric variations. We confirm that the corrected model includes the seasonal change (low-frequency) of steric variations, as it has been corrected by the geoid-referenced observations. Therefore, the volume change related to the density structure is included in the corrected model and the term of dV/dt in this equation also retains the density-related volume changes. We added the following sentence to the paragraph following Figure 3:

*It would be worth noting that the steric effect correction was also included in the corrected DT (Jahanmard et al., 2023a), which may contribute to the seasonal difference.*

Also, L310:

*It should be emphasized that the steric correction is incorporated into the corrected HDM, which as a result, the density-related volume changes are included in the term dV/dt. In addition, the volume change due to land uplift was accounted for through the definition of using absolute DT relative to a reference epoch.*

The steric variation in the Baltic Sea is only 10% of the sea level variation and becomes more significant over the decadal timescale (Virtanen et al., 2010; Karimi et al., 2022). On shorter timescales, the north-south density gradient in the Baltic is almost constant, and the sea level variation in the Baltic Sea is dominated by the barystatic component. The permanent density gradient causes a permanent northward sea surface tilt in this basin (shown in Figure 6a). It is also noteworthy that the water exchange between the Baltic and North Seas is determined using Eq. (5), and river runoff, computed from the volume balance equation, includes steric variations in the corrected model. However, the original model may not account for seasonal steric variation.

References:
Karimi, A. A., Ghobadi-Far, K., & Passaro, M. (2022). Barystatic and steric sea level variations in the Baltic Sea and implications of water exchange with the North Sea in the satellite era. Frontiers in Marine Science, 9, 963564.
Virtanen, J., Mäkinen, J., Bilker-Koivula, M., Virtanen, H., Nordman, M., Kangas, A., ... & Thomas, M. (2010). Baltic sea mass variations from GRACE: comparison with in situ and modelled sea level heights. In Gravity, Geoid and Earth Observation: IAG Commission 2: Gravity Field, Chania, Crete, Greece, 23-27 June 2008 (pp. 571-577). Springer Berlin Heidelberg.

---

## Author Response (AR2)

Revised R2 version, submitted, 05.03.2025
**"Quantification of Baltic Sea Water Budget components Using Dynamic Topography"**
Responses to reviewers' remarks

**REVIEWER #1**

*Please check that all abbreviations are explained, now there is at least NAP (line 102), BSCD2000 (line 140) and E-HYPE (line 148) that are not explained at all.*

Thanks for your observation. The abbriviations are explained in the text. Please refer to lines: 87, 102, 126, and 150